



# Light-induced protein nitration and degradation with HONO emission

Hannah Meusel[1], Yasin Elshorbany[2], Uwe Kuhn[1], Thorsten Bartels-Rausch[3], Kathrin Reinmuth-Selzle[1], Christopher J. Kampf[4], Guo Li[1], Xiaoxiang Wang[1], Jos Lelieveld[5], Ulrich Pöschl[1], Thorsten Hoffmann[6], Hang Su[1,7*], Markus Ammann[3], Yafang Cheng[1,7*]

[1] Max Planck Institute for Chemistry, Multiphase Chemistry Department, Mainz, Germany
[2] NASA Goddard Space Flight Center, Greenbelt, Maryland, USA & Earth System Science Interdisciplinary Center, University of Maryland, College Park, Maryland, USA
[3] Paul Scherer Institute, Villigen, Switzerland
[4] Johannes Gutenberg University, Institute for Organic Chemistry, Mainz, Germany
[5] Max Planck Institute for Chemistry, Atmospheric Chemistry Department, Mainz, Germany
[6] Johannes Gutenberg University, Institute for Inorganic and Analytical Chemistry, Mainz, Germany
[7] Institute for Environmental and Climate Research, Jinan University, Guangzhou, China

* Correspondence to: Y. Cheng (yafang.cheng@mpic.de) or H. Su (h.su@mpic.de)

**Abstract.** Proteins can be nitrated by air pollutants ($NO_2$), enhancing their allergenic potential. This work provides insight into protein nitration and subsequent decomposition in the present of solar radiation. We also investigated light-induced formation of nitrous acid (HONO) from protein surfaces that were nitrated either online with instantaneous gas phase exposure to $NO_2$ or offline by an efficient nitration agent (tetranitromethane, TNM). Bovine serum albumin (BSA) and ovalbumin (OVA) were used as model substances for proteins. Nitration degrees of about 1% were derived applying $NO_2$ concentrations of 100 ppb under VIS/UV illuminated condition, while simultaneous decomposition of (nitrated) proteins was also found during long-term (20h) irradiation exposure. Gas exchange measurements of TNM- nitrated proteins revealed that HONO can be formed and released even without contribution of instantaneous heterogeneous $NO_2$ conversion. However, fumigation with $NO_2$ was found to increase HONO emissions substantially. In particular, a strong dependence of HONO emissions on light intensity, relative humidity (RH), $NO_2$ concentrations and the applied coating thickness were found. The 20 hours long-term studies revealed sustained HONO formation, even if concentrations of the intact (nitrated) proteins were too low to be detected after the gas exchange measurements. A reaction mechanism for the $NO_2$ conversion based on the Langmuir-Hinshelwood kinetics is proposed.

## 1 Introduction

Primary biological aerosols (PBA), or bioaerosols, including proteins, from different sources and with distinct properties, are known to influence atmospheric cloud microphysics and public health (Lang-Yona et al., 2016; D'Amato et al., 2007; Pummer et al., 2015). Bioaerosols represent a diverse subset of atmospheric particulate matter that is directly emitted in form of active or dead organisms, or fragments, like bacteria, fungal spores, pollens, viruses, and plant debris. Proteins are found ubiquitously in the atmosphere as part of these airborne, typically coarse-size biological particles (diameter > 2.5 μm), but also in fine particulate matter (diameter < 2.5 μm) associated with a host of different constituents such as polymers derived from biomaterials and proteins dissolved in





hydrometeors, mixed with fine dust and other particles (Miguel et al. 1999; Riediker et al., 2000; Zhang and
Anastasio, 2003). Proteins contribute up to 5% of particle mass in airborne particles (Franze et al., 2003a; Staton et
al., 2015; Menetrez et al., 2007) and are also found at surfaces of soils and plants. Proteins can be nitrated and are
then likely to enhance allergic responses (Gruijthuijsen et al., 2006). Nitrogen dioxide ($\cdot NO_2$) has emerged as an
important biological reactant and has been shown to be capable of electron (or H atom) abstraction from the amino
acid tyrosine (Tyr) to form TyrO• in aqueous solutions (tyrosine phenoxyl radical, also called tyrosyl radical; Prütz et
al. 1984 and 1985; Alfassi 1987; Houée-Lévin et al., 2015), which subsequently can be nitrated by a second $NO_2$
molecule. Shiraiwa et al. (2012) observed nitration of protein aerosol, but not solely with $NO_2$ in the gasphase, and
demonstrated that simultaneous $O_3$ exposure of airborne proteins in dark conditions can significantly enhance $NO_2$
uptake and consequent protein nitration (3-nitrotyrosine formation) by way of direct $O_3$-mediated formation of the
TyrO• intermediate. A connection between increased allergic diseases and elevated environmental pollution,
especially traffic-related air pollution has been proposed (Ring et al., 2001). Tyrosine is one of the photosensitive
amino acids and it is subject of direct and indirect photo-degradation under solar-simulated conditions (Boreen, et al.,
2008), especially mediated by both UV-B ($\lambda$ 280−320 nm) and UV-A ($\lambda$ 320 −400 nm) radiation (Houee-Levin et al.,
2015; Bensasson et al., 1993). Direct light absorption or absorption by adjacent endogenous or exogenous
chromophores and subsequent energy transfer results in an electronically-excited state of tyrosine (for details see
Houée-Lévin et al. 2015 and references therein). If the triplet state of tyrosine is generated, it can undergo electron
transfer reactions and deprotonation to yield TyrO• (Fig.1, Bensasson 1993; Davies 1991; Berto et al., 2016).
Regardless of how the tyrosyl radical is generated, it can be nitrated by reaction with $NO_2$, but also hydroxylated or
dimerized (Shiraiwa et al., 2012; Reinmuth-Selzle et al., 2014; Kampf et al., 2015).
With respect to atmospheric chemistry, Bejan et al. (2006) have shown that photolysis of ortho-nitrophenols (as is
the case for 3-nitrotyrosine) can generate nitrous acid (HONO). HONO is of great interest for atmospheric
composition, as its photolysis forms OH radicals, being the key oxidant for degradation of most air pollutants in the
troposphere (Levy, 1971). In the lower atmosphere, up to 30% of the primary OH radical production can be
attributed to photolysis of HONO, especially during the early morning when other photochemical OH sources are
still small (R1, Kleffmann et al., 2005; Alicke et al., 2002; Ren et al., 2006; Su et al., 2008; Meusel et al. 2016).

$$HONO \xrightarrow{hv} OH + NO \qquad (hv = 300 - 405 \text{ nm}) \tag{R1}$$

HONO can be directly emitted by combustion of fossil fuel (Kurtenbach et al., 2001) or formed by gas phase
reactions of NO and OH (the backwards reaction of R1) and heterogeneous reactions of $NO_2$ on wet surfaces
according to R2. On carbonaceous surfaces (soot, phenolic compounds) HONO is formed via electron or H transfer
reactions (R3 and R4-R6; Kalberer et al., 1999; Kleffmann et al., 1999; Gutzwiller et al., 2002; Aubin and Abbatt
2007; Han et al., 2013; Arens et al., 2001, 2002; Ammann et al., 1998, 2005).

$$2NO_2 + H_2O \rightarrow HONO + HNO_3 \tag{R2}$$

$$NO_2 + \{C - H\}_{red} \rightarrow HONO + \{C\}_{ox} \tag{R3}$$

$$ArOH + NO_2 \rightarrow ArO \cdot + HONO \tag{R4}$$

$$ArOH + H_2O \rightarrow ArO^- + H_3O^+ \tag{R5}$$

$$ArO^- + NO_2 \rightarrow NO_2^- + ArO \cdot \xrightarrow{H_3O^+} HONO + H_2O \tag{R6}$$



Previous atmospheric measurements and modeling studies have shown unexpected high HONO concentrations
during daytime, which can also contribute to aerosol formation through enhanced oxidation of precursor gases
(Elshorbany et al., 2014). Measured mixing ratios are typically about one order of magnitude higher than simulated
ones, and an additional source of 200-800 ppt h$^{-1}$ would be required to explain observed mixing ratios (Kleffmann et
al., 2005; Acker et al., 2006; Sörgel et al., 2011; Li et al., 2012; Su et al., 2008; Elshorbany et al., 2012; Meusel et al.,
2016) indicating that estimates of daytime HONO sources are still under debate. It was suggested that HONO arises
from the photolysis of nitric acid and nitrate or by heterogeneous photochemistry of $NO_2$ on organic substrates and
soot (Zhou et al., 2001; 2002 and 2003; Villena et al., 2011; Ramazan et al., 2004; George et al., 2005; Sosedova et
al., 2011; Monge et al., 2010; Han et al., 2016). Stemmler et al. (2006, 2007) found HONO formation on light-
activated humic acid, and field studies showed that HONO formation correlates with aerosol surface area, $NO_2$ and
solar radiation (Su et al., 2008; Reisinger, 2000; Costabile et al., 2010; Wong et al., 2012; Sörgel et al., 2015) and is
increased during foggy periods (Notholt et al., 1992). Another proposed source of HONO is the soil, where it has
been found to be co-emitted with NO by soil biological activities (Oswald et al., 2013; Su et al., 2011; Weber et al.,
14 2015).

In view of light-induced nitration of proteins and HONO formation by photolysis of nitro-phenols, light-enhanced
production of HONO on protein surfaces can be anticipated, which, to the best of our knowledge, has not been
studied before.
This work aims at providing insight into protein nitration, the atmospheric stability of the nitrated protein, and
respective formation of HONO from protein surfaces that were nitrated either offline in liquid phase prior to the gas
exchange measurements, or online with instantaneous gas phase exposure to $NO_2$, with particular emphasis on
environmental parameters like light intensity, relative humidity (RH) und $NO_2$ concentrations. Bovine serum
albumin (BSA), a globular protein with a molecular mass of 66.5 kDa and 21 tyrosine residues per molecule, was
chosen as a well-defined model substance for proteins. Nitrated ovalbumin (OVA) was used to study the light-
induced degradation of proteins that were nitrated prior to gas exchange measurements. This well-studied protein has
a molecular mass of 45 kDa and 10 tyrosine residues per molecule.
**2 Materials and methods**
**2.1 Protein preparation and analysis**
BSA (albumin from bovine serum, Cohn V fraction, lyophilized powder, ≥ 96%; Sigma Aldrich, St. Louis, Missouri,
USA) or nitrated OVA (ovalbumin) was solved in pure water (18.2MΩ cm) and coated onto the glass tube.
The nitration of ovalbumin (OVA) was described previously (Yang et al., 2010; Zhang et al., 2011). Briefly, OVA
(Grade V, A5503-5G, Sigma Aldrich, Germany) was dissolved in phosphate buffered saline PBS (P4417-50TAB,
Sigma Aldrich, Germany) to a concentration of 10 mg/ml. 50 µl tetranitromethane TNM (T25003-5G, Sigma
Aldrich, Germany) dissolved in methanol 4% (v/v) were added to a 2.5 ml aliquot of the OVA solution and stirred
for 180 min at room temperature. Size exclusion chromatography columns (PD-10 Sephadex G-25 M, 17-0851-01,
GE Healthcare, Germany) were used for clean-up. The eluate was dried in a freeze dryer and stored in a refrigerator
at 4°C.





After the flow-tube-experiments (see below) the proteins were extracted with water from the tube and analyzed with
liquid chromatography (HPLC-DAD; Agilent Technologies 1200 series) according to Selzle et al. (2013). This
method provides a straightforward and efficient way to determine the nitration of proteins. Briefly, a monomerically
bound C18 column (Vydac 238TP, 250 mm×2.1 mm inner diameter, 5 μm particle size; Grace Vydac, Alltech) was
used for chromatographic separation. Eluents were 0.1 % (v/v) trifluoroacetic acid in water (LiChrosolv) (eluent A)
and acetonitrile (ROTISOLV HPLC Gradient Grade, Carl Roth GmbH + Co. KG, Germany) (eluent B). Gradient
elution was performed at a flow rate of 200 μL/min. ChemStation software (Rev. B.03.01, Agilent) was used for
system control and data analysis. For each chromatographic run, the solvent gradient started at 3% B followed by a
linear gradient to 90% B within 15 min, flushing back to 3% B within 0.2 min, and maintaining 3% B for additional
2.8 min. Column re-equilibration time was 5 min before the next run. Absorbance was monitored at wavelengths of
280 and 357 nm. The sample injection volume was 10-30 μL. Each chromatographic run was repeated three times.
The protein nitration degree was determined by the method of Selzle et al. (2013). Native and un-treated BSA did not
show any degree of nitration.
**2.2 Coated-wall flow tube system**
Figure 2 shows a flowchart of the set-up of the experiment. $NO_2$ was provided in a gas bottle (1 ppm in $N_2$, Carbagas
AG, Grümligen, Switzerland). $NO_2$ was further diluted (mass flow controller, MFC3) with humidified pure nitrogen
to achieve $NO_2$ mixing ratios between 20 and 100 ppb. Impurities of HONO in the $NO_2$-gas cylinder were removed
by means of a HONO scrubber. The $Na_2CO_3$ trap was prepared by soaking 4mm firebrick in a saturated $Na_2CO_3$ in
50% ethanol / water solution and drying for 24 hours. The impregnated firebrick granules were put into a 0.8 cm
inner diameter and 15 cm long glass tube, which was closed by quartz wool plugs on both sides. A constant total
flow was provided by means of another $N_2$ mass flow controller (MFC2) that compensated for changes in $NO_2$
addition. Different fractions of total surface areas (50, 70 and 100%) of the reaction tube (50 cm x 0.81 cm i.d.) were
coated with 2 mg BSA or nitrated OVA, respectively. Therefore 2 mg protein was dissolved in 600 μL pure water,
injected into the tube and then gently dried in a low humidity $N_2$ flow (RH ~ 30-40%) with continuous rotation of the
tube. The coated reaction tube was exposed to the generated gas mixture and irradiated with either (i) 1, 3 or 7 VIS
lights (400-700 nm; L 15 W/954, lumilux de luxe daylight, Osram, Augsburg, Germany) which is 0, 23, 69 or 161 W
$m^{-2}$ respectively or (ii) 4 VIS and 3 UV lights (340-400 nm; UV-A, TL-D 15 W/10, Philips, Hamburg, Germany).
An overview of the experiments performed during this study is shown in table 1. Light induced decomposition of
nitrated proteins was studied on OVA. Instantaneous $NO_2$ transformation and its light- and RH- dependence on
heterogeneous HONO formation were studied on BSA in short-term experiments. Extended studies on BSA were
performed to explore the persistence of the surface reactivity and respective catalytic effects.
A commercial long path absorption photometry instrument (LOPAP, QUMA) was used for HONO analysis. The
measurement technique was introduced by Heland et al. (2001). This wet chemical analytical method has an
unmatched low detection limit of 3-5 ppt with high HONO collection efficiency (≥ 99%). HONO is continuously
trapped in a stripping coil flushed with an acidic solution of sulfanilamide. In a second reaction with n-(1-
naphthyl)ethylenediamine-dihydrochloride an azo dye is formed, whose concentration is determined by absorption
photometry in a long Teflon tubing. LOPAP has two stripping coils in series to reduce known interferences. In the





first stripping coil HONO is quantitatively collected. Due to the acidic stripping solution, interfering species are
collected less efficiently but in both channels. The true concentration of HONO is obtained by subtracting the
interferences quantified in the second channel from the total signal obtained in the first channel. The accuracy of the
HONO measurements was 10%, based on the uncertainties of liquid and gas flow, concentration of calibration
standard and regression of calibration.
The reagents were all high-purity-grade chemicals, i.e., hydrochloric acid (37 %, ACS reagent, Sigma Aldrich, St.
Louis, Missouri, USA), sulfanilamide (for analysis, >99 %; Sigma Aldrich) and N-(1-naphthyl)-ethylenediamine
dihydrochloride (>98%; ACS reagent, Fluka by Sigma Aldrich). For calibration Titrisol® 1000 mg $NO_2^-$ (NaNO$_2$ in
H$_2$O; Merck) was diluted to 0.001 mg/L $NO_2^-$. For preparation of all solutions and for cleaning of the absorption
tubes 18MΩ H$_2$O was used.
NO$_x$ concentrations were analyzed by means of a commercial chemiluminescence detector from EcoPhysics (CLD
AM, Duernten, Switzerland).
**3 Results and discussion**
**3.1 BSA nitration and degradation**
Nitrated proteins can lead to a stronger allergic response. Nitration of proteins can be enhanced by O$_3$ activation (in
the dark). In the environment, about half a day light is present. What happens with irradiated proteins when exposed
to NO$_2$. Can they be nitrated efficiently? To investigate the degree of protein nitration under illuminated conditions,
BSA coated on the reaction tube (17.5 µg cm$^{-2}$) was exposed to 7 VIS lamps (40% of a clear sky irradiance for a
solar zenith of 48°; Stemmler et al., 2006) and 100 ppb NO$_2$ at 70% RH. After 20 hours the BSA nitration degree
(ND, concentration of nitrated tyrosine residues divided by the total concentration of tyrosine residues) investigated
by means of the HPLC-DAD method was (1.0 ± 0.1)%. Introducing UV radiation (4 VIS plus 3 UV lamps) resulted
in a slightly higher ND of (1.1 ± 0.1)%. Note that no intact protein could be detected by HPLC-DAD after another 20
hours of irradiation without NO$_2$, indicating light induced decomposition of proteins. However, the applied HPLC-
DAD technique only detects (nitro-)tyrosine residues in proteins, and does not provide information about protein
fragments or single nitrated or non-nitrated tyrosine residues. Hence, proteins might have been decomposed while
tyrosine remains in its nitrated form, not detectable by our analysis method. Similarly, proteins (here: OVA) that
were nitrated with TNM in aqueous phase prior to coating (21.5 µg cm$^{-2}$) to an extent of 12.5% also decomposed
when illuminated about 6 hours (1-7 VIS lights; with and without 20 ppb NO$_2$). Thus the nitration of proteins by
light and NO$_2$ was confirmed, but with simultaneous gradual decomposition of the proteins. Effects of UV irradiation
(240-340 nm) on proteins containing aromatic amino acids were reviewed previously (Neves-Peterson et al., 2012).
It was shown that triplet state tryptophan and tyrosine can transfer electron to a nearby disulfide bridge to form the
tryptophan and tyrosine radical. The disulfide bridge could break leading to conformational changes in the protein
but not necessarily resulting in inactivation of the protein. In strong UV light (≈200 nm) the peptide bond could also
break (Nikogosyan and Görner, 1999).
Franze et al. (2005) analyzed a variety of natural samples (road dust, window dust and particulate matter PM 2.5)
collected in the metropolitan area of Munich, containing 0.08-21 g/kg proteins, and revealed equivalent degrees of




nitration (EDN, concentration of nitrated protein divided by concentration of all proteins) between 0.01 and 0.1%
only. Such low nitration degree is in line with light induced decomposition of (nitrated) proteins. On the other hand,
an EDN up to 10% (average 5%) was found for BSA and birch pollen extract (BPE) exposed to Munich ambient air
for two weeks under dark conditions, with daily mean $NO_2$ ($O_3$) concentration of 17 to 50 ppb (7 to 43 ppb) in the
same study, suggesting the deficiency of decomposition without being irradiated. BSA and OVA loaded on syringe-
filters and exposed to 200 ppb $NO_2/O_3$ for 6 days under dark conditions were nitrated to 6 and 8%, respectively
(Yang et al., 2010). Reinmuth-Selzle et al. (2014) found similar ND for major birch pollen allergen Bet v 1 loaded on
syringe-filters exposed to 80-470 ppb $NO_2$ and $O_3$. When exposed for 3-72 hours to $NO_2/O_3$ at RH < 92% the ND
was 2-4%, while at condensing conditions (RH > 98%) the ND increased to 6% after less than one day (19 hours).
The ND of Bet v 1 was considerably increased to 22% for proteins solved in the aqueous phase (0.16 mg mL$^{-1}$) when
bubbling with a 120 ppb $NO_2/O_3$ gas mixture for a similar period of time (17 hours). Other nitration methods,
investigated by Reinmuth-Selzle et al. (2014), e.g., nitration of Bet v 1 with peroxynitrite (ONOO$^-$, formed by
reaction of NO with $O_2^-$) or TNM lead to ND between 10 and 72% depending on reaction time, reagent concentration
and temperature. Similarly high NDs of 45-50% were obtained by aqueous phase TNM nitration of BSA and OVA
by Yang et al. (2010).
**3.2 HONO formation**
**3.2.1 HONO formation from nitrated proteins**
Strong HONO emissions were found for OVA nitrated in the liquid phase prior to gas exchange measurements (ND
= 12.5%). A strong correlation between HONO emission and light intensity was observed (50% RH; Fig. 3). Initially,
we did not apply $NO_2$. Thus the observed HONO formation (up to 950 ppt) originated from decomposing nitrated
proteins rather than from heterogeneous conversion of $NO_2$. However, when exposed to 20 ppb of $NO_2$ in dark
conditions, HONO formation increased 4-fold (50 to 200 ppt), and about 2-fold with 7 VIS lamps turned on (950 to
1800 ppt). After 7 hours of flow tube experiments (4.5 h irradiation with varying light intensities (0-1-3-7 lights) +
2.5 h irradiation/20 ppb $NO_2$ (7-3-0- lights)), no intact protein was found according to the analysis of HPLC-DAD.
**3.2.2 Light dependency**
To investigate HONO formation on unmodified BSA coating (31.4 μg cm$^{-2}$) in dependence on light conditions, the
radiation intensity (number of VIS lamps) was changed under otherwise constant conditions of exposure at 20 ppb
$NO_2$ and 50% RH. Decreasing light intensity revealed a linearly decreasing trend in HONO formation from about
1000 ppt to 140 ppt (red symbols in Fig. 4). After re-illumination to the initial high light intensity the HONO
formation was reduced by 32% (blue symbol in Fig. 4). Stemmler et al. (2006) and Sosedova et al. (2011) also
observed a similar saturation of HONO formation on humic acid, tannic and gentisic acid at higher light intensities.
Stemmler et al. (2006) argued that surface sites activated for $NO_2$ heterogeneous conversion by light (R3) would
become de-activated by competition with photo-induced oxidants (X$^*$, R7-8), e.g., primary chromophores or electron
donors are oxidized by surface*, which is in line with the observed decomposition of the native protein presented
above.





$$surface \xrightarrow{hv} surface^* \xrightarrow{NO_2} HONO + surface_{ox} \qquad \text{(R7)}$$

$$X \xrightarrow{hv} X^* \xrightarrow{surface^*} surface - X \qquad \text{(R8)}$$

In other studies the $NO_2$ uptake coefficient on soot, mineral dust, humic acid and other solid organic compounds
similarly increased at increasing light intensities (George et al., 2005; Stemmler et al., 2007; Ndour et al., 2008;
Monge et al., 2010; Han et al., 2016). Note that the HONO yield (ratio of HONO formed to $NO_2$ lost) was found to
be constant at light intensities in the range of 60-200 W m$^{-2}$ in the work of Han et al. (2016), but have shown a linear
dependence on light for nitrated phenols (Bejan et al., 2006).

### 3.2.3 NO₂ dependency

At about 50% relative humidity and high illumination intensities (7 VIS lamps, ~161 W m$^{-2}$), heterogeneous
formation of HONO strongly correlated with the applied $NO_2$ concentration (Fig. 5). On a BSA surface of about 16.1
μg cm$^{-2}$ (Tab. 1) the produced HONO concentration increased from 56 ppt at 20 ppb $NO_2$ to 160 ppt at 100 ppb $NO_2$.
Only at a threshold $NO_2$ level well above those typically observed in natural environments (>>150 ppb) this
increasing trend slowed down to some extent, indicative of saturation of active surface sites. A similar pattern of
$NO_2$ dependence was also observed for light-induced HONO formation from humic acid (Stemmler et al., 2006) and
phenolic compounds like gentisic and tannic acid (Sosedova et al., 2011), and for heterogeneous $NO_2$ conversion on
soot under dark conditions (Stadler and Rossi, 2000; Salgado and Rossi, 2002; Arens et al., 2001).
For better comparison of the different studies the HONO concentration measured at different $NO_2$ concentrations
was normalized to the HONO concentration at 20 ppb $NO_2$ ([HONO]$_{NO2}$/[HONO]$_{NO2=20ppb}$) in Fig. 5, as variable
absolute amounts of HONO were found in different studies and matrices. A cease of the $NO_2$ dependency on
heterogeneous HONO formation can be assessed for most of the studies at $NO_2$ concentrations $\geq$ 200 ppb. A very
similar correlation (up to 40 ppb $NO_2$) was observed when $NO_2$ was applied additionally during the gas phase
photolysis of nitrophenols (fig. 5; Bejan et al., 2006). Even though the matrix (nitrophenols) and conditions
(illuminated) of the latter is comparable to the experiment presented here, for BSA no clear indication of saturation
was found up to 160 ppb of $NO_2$, pointing to a highly reactive surface of BSA for $NO_2$ under illuminated conditions.
As shown with R7 and R8, the concentration dependence depends on the competing channel R8, therefore, this is
strongly matrix dependent, both in terms of chemical and physical properties.

### 3.2.4 Impact of coating thickness

Strong differences in HONO concentrations were found for experiments with different coating thicknesses applying
otherwise similar conditions (20 ppb of $NO_2$, 7 VIS lamps and 50% RH). While only 55 ppt of HONO concentration
was observed for a shallow homogeneous coating of 16.1 μg cm$^{-2}$ (217.6 nm thickness, see below) applied on the
whole length of the tube, up to 2 ppb were found for a thick (more uneven) coating of 31.44 μg cm$^{-2}$ (435.2 nm
thickness) covering only 50% of the tube (Fig. 6). Potential explanations are that thicker coating leads to (1) more
bulk reactions producing HONO, or (2) different morphologies, e.g., higher effective reaction surfaces.
A strong increase in $NO_2$ uptake coefficients with increasing coating thickness was also observed for humic acid
coatings (Han et al., 2016). However, they found an upper threshold value of 2 μg cm$^{-2}$ of cover load (20 nm





absolute thickness, assuming a humic acid density of 1 g cm$^{-3}$), above which uptake coefficients were found to be
constant. The authors also proposed that NO$_2$ can diffuse deeper into the coating and below 2 µg cm$^{-2}$ the full cover
depth would react with NO$_2$, respectively.
For proteins the number of molecules per monolayer depends on their orientation and respective layer thickness can
vary accordingly. One (dry, crystalline) BSA molecule has a volume of about 154 nm$^3$ (Bujacz, 2012). In a flat
orientation (4.4 nm layer height, and a projecting area of 35 nm² per molecule) 3.64x10$^{14}$ molecules (40.5 µg; 0.32
µg cm$^{-2}$) of BSA are needed to form one complete monolayer in the flow tube (i.d. of 0.81 cm, 50 cm length, 100%
surface coating). Hence, the thinnest BSA coating applied in the experiment (16.1 µg cm$^{-2}$) would consist of 50
monolayers revealing a total coating thickness of 217.6 nm, and the thickest BSA coating (31 µg cm$^{-2}$) would have
99 monolayers and an absolute thickness of 435.1 nm. At the other extreme (non-flat) orientation, more BSA
molecules are needed to sustain one monolayer. With 21.7 nm² of projected area of one molecule and 7.1 nm
monolayer height, 5.86x10$^{14}$ molecules of BSA are needed to form one complete monolayer in the flow tube. The
coatings would consist of between 31 (thinnest) and 61 (thickest) monolayers of BSA. With a flat orientation 1-2%
(number or weight) of BSA molecules would build the uppermost surface monolayer, whereas in an upright
molecule orientation 1.6-3.3% would be in direct contact with surface ambient air.
In the crystalline form several molecules of water stick tightly to BSA. As BSA is highly hygroscopic, more water
molecules are adsorbed at higher relative humidity. At 35% RH BSA is deliquesced (Mikhailov et al., 2004).
Therefore the above described number of monolayers and the absolute layer thickness are a lower bound estimate.
Conclusively, the thickness dependence on HONO formation is extremely complex. Activation and photolysis of
nitrated Tyr occurs throughout the BSA layer. The heterogeneous reaction of NO$_2$ may or may be not limited to the
surface depending on solubility and diffusivity of NO$_2$. Also the release of HONO may be limited by diffusion.
**3.2.5 RH dependency**
The dependence of HONO emission on relative humidity is shown in Fig. 7. Here about 25 ppb of NO$_2$ was applied
to a (not nitrated) BSA coated flow tube (17.5 µg cm$^{-2}$) both in dark and illuminated conditions (7 VIS lights).
HONO formation scaled with relative humidity. Kleffmann et al. (1999) proposed that higher humidity inhibits the
self-reaction of HONO (2 HONO$_{(s, g)}$ → NO$_2$ + NO + H$_2$O), which leads to higher HONO yield from heterogeneous
NO$_2$ conversion.
The RH dependence of HONO formation on proteins is different to other surfaces. For example, no influence of RH
has been observed for dark heterogeneous HONO formation on soot particles sampled on filters (Arens et al., 2001).
For HONO formation on tannic acid coatings (both at dark and irradiated conditions) a linear but relatively weak
dependence has been reported between 10 and 60% RH, while below 10% and above 60% RH the correlation
between HONO formation and RH was much stronger (Sosedova et al., 2011). Similar results were observed for
anthrarobin coatings by Arens et al. (2002). This type of dependence of HONO formation on phenolic surfaces on
RH equals the HONO formation on glass, following the BET water uptake isotherm of water on polar surfaces
(Finnlayson-Pitts et al., 2003; Summer et al., 2004). For humic acid surfaces the NO$_2$ uptake coefficients also weakly
increased below 20% RH and were found to be constant between 20 and 60% (Stemmler et al., 2007).



While on solid matter chemical reactions are essentially confined to the surface rather than in the bulk, proteins can
adopt an amorphous solid or semisolid state, influencing the rate of heterogeneous reactions and multiphase
processes. Molecular diffusion in the non-solid phase affects the gas uptake and respective chemical transformation.
Shiraiwa et al. (2011) could show that the ozonolysis of amorphous protein is kinetically limited by bulk diffusion.
The reactive gas uptake exhibits a pronounced increase with relative humidity, which can be explained by a decrease
of viscosity and increase of diffusivity, as the uptake of water transforms the amorphous organic matrix from a
glassy to a semisolid state (moisture-induced phase transition). The viscosity and diffusivity of proteins depend
strongly on the ambient relative humidity because water can act as a plasticizer and increase the mobility of the
protein matrix (for details see Shiraiwa et al. 2011 and references therein). Shiraiwa et al. (2011) further showed that
the BSA phase changes from solid through semisolid to viscous liquid as RH increases, while trace gas diffusion
coefficients increased about 10 orders of magnitude. This way, characteristic times for heterogeneous reaction rates
can decrease from seconds to days as the rate of diffusion in semisolid phases can decrease by multiple orders of
magnitude in response to both low temperature (not investigated in here) and/or low relative humidity. Accordingly,
we propose that HONO formation rate depends on the condensed phase diffusion coefficients of $NO_2$ diffusing into
the protein bulk, HONO released from the bulk and mobility of excited intermediates.
**3.2.6 Long term exposure with $NO_2$ under irradiated conditions**
To study long-term effects of irradiation on HONO formation from proteins, flow tubes were coated with 2 mg BSA
($17.5 \pm 0.4$ μg cm$^{-2}$; 90% of total length) and exposed to 100 ppb $NO_2$, at 80% RH at illuminated conditions for a
time period of up to 20 hours (Fig. 8). Samples illuminated with VIS light only (red and orange colored lines in Fig.
8) showed persistent HONO emissions over the whole measurement period. For reasons unknown, and even though
the observed HONO concentrations were within the expected range with regard to the applied $NO_2$ concentrations,
RH and cover characteristics, one sample (orange in Fig. 8) showed a sharp short-term increase in the initial phase
followed by respective decrease, not in line with all other samples (compare Fig. 6). However, after 4 hours both VIS
irradiated samples showed virtually constant HONO emissions (-3.8 and +1.6 ppt h$^{-1}$, respectively). The sample
illuminated with UV/VIS light (3 UV and 4 VIS lamps) showed a sustained sharp increase in the first 4 hours,
followed by persistent and very stable (decay rate as low as -0.5 ppt h$^{-1}$) HONO emissions at an about 3-fold higher
level compared to samples irradiated with VIS only.
Integrating the 20 hour experiments, $9.23 \times 10^{15}$ (4.6 ppb*h, VISa), $1.53 \times 10^{16}$ (7.7 ppb*h, VISb) and $4.01 \times 10^{16}$ (20
ppb*h, UV/VIS) molecules of HONO were produced. This means between $7.7 \times 10^{13}$ and $3.3 \times 10^{14}$ molecules of
HONO per cm$^2$ of BSA geometric surface were formed. With respect to the different experimental conditions
concerning cover thickness, RH, and $NO_2$ concentrations, this is in a similar order of magnitude as found for humic
acid ($2 \times 10^{15}$ molecules cm$^{-2}$ in 13 hours) by Stemmler et al. (2006).
If BSA acts like a catalytic converter as in a Langmuir-Hinshelwood reaction each BSA molecule can react several
times with $NO_2$ to heterogeneously form HONO. As described in 3.1, BSA nitration is in competition with $NO_2$
surface reactions and only a limited number of $NO_2$ molecules could react with BSA forming HONO via nitration of
proteins and subsequent decomposition of nitrated proteins. A BSA molecule contains 21 tyrosine residues, which
could react with $NO_2$. But even a strong nitration agent such as TNM is not capable of nitrating all tyrosine residues



and a mean nitration degree of 19% was found (Peterson et al., 2001; Yang et al., 2010), i.e., 4 tyrosine residues of
one BSA molecule can be nitrated to form HONO. As 2 mg of BSA was applied for each flow tube coating, a total of
$1.8 \times 10^{16}$ protein molecules can be inferred. In 20 hours of irradiating with VIS light 13-22% of the accessible Tyr
residues (4 each BSA molecule) would have been reacted. Irradiating with additional UV lights at least 56% of the
tyrosine residues would have been nitrated and decomposed, respectively. But as $NO_2$ is a much weaker nitrating
agent and nitration of only one tyrosine residue is probable (ND of BSA with $O_3/NO_2$ 6%; Yang et al., 2010) up to
85% BSA molecules would have been reacted when irradiated with VIS lights, and even more HONO molecules as
coated BSA molecules would have been generated under UV/VIS light conditions. Other amino-acids of the protein
like tryptophan or phenylalanine might also be nitrated but without formation of HONO (Goeschen et al., 2011).
Hence, a contribution of heterogeneous conversion of $NO_2$ can be anticipated.
**3.3 Kinetic studies**
The experimental results (especially the stability over a long time) indicate that the formation of HONO from $NO_2$
on protein surfaces likely underlies the Langmuir-Hinshelwood mechanism in which the protein would act as a
catalytic converter (Fig. 9). The first step is the fast reversible physical adsorption of $NO_2$ ($k_1$) and water followed by
the slow conversion into HONO (eq.1 and eq.2). In our experiments and in the atmosphere there is always sufficient
water and for simplification we assume that the reaction rate only depends on $NO_2$.

$$\frac{d[NO_2]_s}{dt} = k_1 * [NO_2]_g \qquad \text{(eq.1)}$$

$$\frac{d[HONO]_s^1}{dt} = k_2 * [NO_2]_s \qquad \text{(eq.2)}$$

where index s and g indicate sorbed and gaseous state, respectively.
From the experiments in which higher HONO concentrations were detected with higher light intensities we conclude
that the heterogeneous conversion of $NO_2$ to HONO is light induced or a photochemical reaction. It was observed
that the nitration of proteins is a competitive (side) reaction of the direct HONO formation (eq.2) but light induced
decomposition of nitrated protein also produces HONO (eq.3).
$$\frac{d[HONO]_s^2}{dt} = k_4 * k_5 * [NO_2]_s \qquad \text{(eq.3)}$$

As these two processes cannot be discriminated by the observations presented here, we combine both reactions to
formulate an overall formation equation (eq.4) with k' = $k_2 + k_4*k_5$
$$\frac{d[HONO]_s}{dt} = [HONO]_s^1 + [HONO]_s^2 = k' * [NO_2]_s \qquad \text{(eq.4)}$$

The final step of the mechanism is the release of the generated HONO into the air. Since proteins are in general
slightly acidic, the desorption of HONO ($k_3$) should be fairly fast (eq.5).
$$\frac{d[HONO]_g}{dt} = k_3 * [HONO]_s \qquad \text{(eq.5)}$$

An effective formation rate of gaseous $NO_2$ to gaseous HONO $k_{eff}$ was calculated according to eq.6.
$$\frac{d[HONO]_g}{dt} = k_{eff} * [NO_2]_g \qquad \text{(eq.6)}$$

with $k_{eff} = k_1*k'*k_3$
In the first 5-10 min of the long-term experiments HONO increased (Fig. 8 – zoomed in range). This slope was taken
as $d[HONO]_g/dt$ in eq.6. Effective rate constants between $1.48 \times 10^{-6}$ s$^{-1}$ (VIS a) and $7.40 \times 10^{-6}$ s$^{-1}$ (VIS b) were



calculated. When irradiating with VIS light only, the concentration of HONO was either constant or decreased for 2
h after this first 10 min. When irradiating with additional UV light, the HONO signal showed an enhancement in two
steps. In the first 10 min it was strongly increasing (1327 ppt $h^{-1}$) and then in the next hour it increased less with 170
ppt $h^{-1}$ prior to stabilization. Therefore two rate constants of $4.10 \times 10^{-6}$ $s^{-1}$ and $5.2 \times 10^{-7}$ $s^{-1}$ were obtained, respectively.
Reactive uptake coefficients for $NO_2$ were calculated according to Li et al. (2016). For both irradiation types the
uptake coefficient $\gamma$ was in the range of $7 \times 10^{-6}$ at the very beginning of each experiment. After a few minutes they
decreased to a mean of $1 \times 10^{-7}$. The calculated $k_{eff}$ values and uptake coefficient are in the same range and match the
$NO_2$ uptake coefficients on irradiated humic acid surfaces (coatings) and aerosols obtained by Stemmler et al. (2006
and 2007) which were in between $2 \times 10^{-6}$ and $2 \times 10^{-5}$ (coatings) and $1 \times 10^{-6}$ and $6 \times 10^{-6}$ (aerosols), depending on $NO_2$
concentrations and light intensities. Similar $NO_2$ uptake coefficients on humic acid were observed by Han et al.
(2016). George et al., (2005) reported about a two-fold increased $NO_2$ uptake coefficients for irradiated organic
substrates (benzophenone, catechol, anthracene) compared to dark conditions, in the order of $(0.6-5) \times 10^{-6}$. $NO_2$
uptake coefficients on gentisic acid and tannic acid were in between $(3.3-4.8) \times 10^{-7}$ (Sosedova et al., 2011), still
being higher than on fresh soot or dust (about $1 \times 10^{-7}$; Monge et al., 2010; Ndour et al., 2008). The $NO_2$ uptake
coefficients on BSA in presence of $O_3$ ($1 \times 10^{-5}$, for 26 ppb $NO_2$ and 20 ppb $O_3$) published by Shiraiwa et al. (2012)
were somewhat higher than the values calculated here without $O_3$ but with light.
As proteins can efficiently be nitrated by $O_3$ and $NO_2$ in polluted air (Franze et al., 2005, Shiraiwa et al., 2012;
Reinmuth-Selzle et al. 2014), the emission of HONO from light-induced decomposing nitrated proteins could play an
important role in the HONO budget. As proteins are nitrated at their tyrosine residues (at the ortho position to the OH
group on the aromatic ring) the underlying mechanism of this HONO formation should be very similar to the HONO
formation by photolysis of ortho-nitrophenols described by Bejan et al. (2006). This starts with a photo-induced
hydrogen transfer from the OH group to the vicinal $NO_2$ group (Fig. 1), which leads to an excited intermediate from
which HONO is eliminated subsequently.
**4. Summary and Conclusion**
Photochemical nitration of proteins accompanied by formation of HONO by (i) heterogeneous conversion of $NO_2$
and (ii) by decomposition of nitrated proteins was studied under relevant atmospheric conditions. $NO_2$ concentrations
ranged from 20 ppb (typical for urban regions in Europe and USA) up to 100 ppb (representative for highly polluted
industrial regions). The applied relative humidity of up to 80% and light intensities of up to 161 $W/m^2$ are common
on cloudy days. Under illuminated conditions very low nitration of proteins or even no native protein was observed,
indicating a light-induced decomposition of nitrated proteins to shorter peptides. These might still include nitrated
residues of which potential health effects are not yet known. An average effective rate constant of the total $NO_2$-
HONO conversion of $3.3 \times 10^{-6}$ $s^{-1}$ (for about 120 $cm^2$ of protein surface and a layer volume of 0.003 $cm^3$;
surface/volume ratio ~ 40000 $cm^{-1}$) was obtained. At 20 ppb $NO_2$ 238 ppt $h^{-1}$ HONO would be formed. Projecting
this to $1m^2$ of pure BSA surface a formation of 19.8 ppb HONO $h^{-1}$ $m^{-2}$ could be estimated. No data about
representative protein surface areas on atmospheric aerosol particles are available. However, the number and mass
concentration of primary biological aerosol particles such as pollen, fungal spores and bacteria, containing proteins,


are in the range of 10-10$^4$ m$^{-3}$ and 10$^{-3}$-1 µg m$^{-3}$, respectively (Shiraiwa et al., 2012). Therefore it is difficult to
estimate the importance of HONO formation on protein surface and its contribution to the HONO budget. In many
studies the calculated un-known source strength of daytime HONO formation is with a range of about 200-800 ppt
h$^{-1}$ (Kleffmann et al., 2005; Acker et al., 2006; Li et al., 2012).
**Acknowledgment**
This study was supported by the Max Planck Society (MPG).

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





**Tables and Figures**
**Tab 1: Details on the different experiments, aims and experimental conditions (coating, applied NO$_2$ concentration,**
**number of lights switched on, relative humidity and time for each exposure step):**

| | | Coating density (number of monolayers NML$_f$, thickness) | NO$_2$ [ppb] | no. of lamps | RH [%] | time per step [h] |
|---|---|---|---|---|---|---|
| **A** | *light induced decomposition of nitrated protein and HONO formation* | | | | | |
| 1 | light and NO$_2$ dependency | n-OVA 21.5 ± 0.8 µg cm$^{-2}$ (68 NML$_f$ , 298.05 nm) | 0-20 | 0-1-3-7 VIS | 50 | 1 |
| **B** | *heterogeneous NO$_2$ transformation on BSA* | | | | | |
| 2 | NO$_2$ dependency | BSA 16.1±0.4 µg cm$^{-2}$ (50 NML$_f$ , 217.6 nm) | 0-20-40-60-100 | 7 VIS | 50 | 0.5-1 |
| 3 | light dependency | BSA 31.4±1.4 µg cm$^{-2}$ (99 NML$_f$ , 435.2 nm) | 20 | 0-1-3-7 VIS | 50 | 0.5-1 |
| 4 | coating thickness | BSA 16.1±0.4 µg cm$^{-2}$ (50 NML$_f$ , 217.6 nm), 22.5±0.8 µg cm$^{-2}$ (71 NML$_f$ , 310.8 nm), 31.4±1.4 µg cm$^{-2}$ (99 NML$_f$, 435.2 nm) | 20 | 7 VIS | | 0.5-3 |
| 5 | RH dependency | BSA 17.5±0.4 µg cm$^{-2}$ (55 NML$_f$, 241.7 nm) | 25 | 0-7VIS | 0-50-80 | 0.25-1 |
| 6 | time effect | BSA 17.5±0.4 µg cm$^{-2}$ | 100 | 7 VIS | 75 | 20 |
| 7 | time effect | BSA 17.5±0.4 µg cm$^{-2}$ | 100 | 4 VIS + 3 UV | 75 | 20 |

NML$_f$ numbers of monolayers in flat orientation

**Fig. 1: Reaction mechanism of atmospheric BSA nitration and subsequent HONO emission (formation of the tyrosine**
**phenoxyl radical and following NO$_2$ addition to 3-nitrotyrosine was adapted from Houée-Levin et al. (2015) and Shiraiwa**
**et al. (2012); intramolecular H-transfer adapted from Bejan et al., 2006).**





**Fig. 2: flow system and set-up, MFC = mass flow controller**

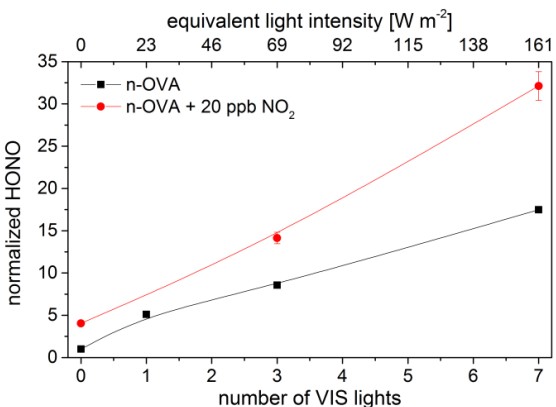

**Fig. 3: Light enhanced HONO formation from proteins nitrated in the liquid phase prior to the flow tube experiments (n-**
**OVA: ND 12.5%, coating 21.5 µg cm$^{-2}$) with and without additional NO$_2$ in the purging air at 50% RH (HONO is**
**normalized to the HONO concentration measured without NO$_2$ and no light ([HONO]$_{lights; NO2}$/[HONO]$_{dark; NO2=0}$)) .**



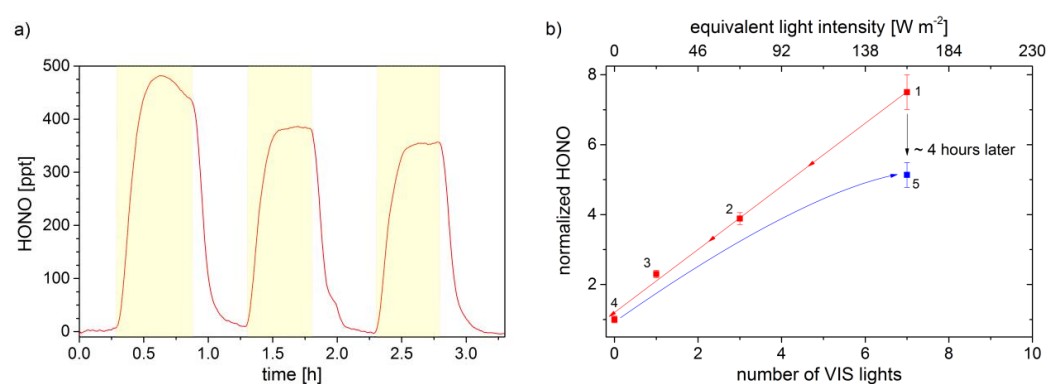

**Fig. 4: a) Light enhancement of HONO formation on BSA surface (22.5 µg cm$^{-2}$), yellow shaded areas indicate periods in**
**which 7 VIS lamps were switched on (RH = 50%, NO$_2$ = 20 ppb); b) Dependency of HONO formation on radiation**
**intensity at 20 ppb NO$_2$ and 50% RH (BSA = 31.4 µg cm$^{-2}$). The experiment started with 7 VIS lights switched on,**
**sequentially decreasing the number of lights (red symbols, nominated 1-4), prior to apply the initial irradiance again (blue**
**symbol, 5). HONO was normalized to the HONO concentration in darkness ([HONO]$_{lights}$/[HONO]$_{dark}$). Error bars**
**indicate standard deviation of 20-30 min measurements, standard deviation of point 5 covers 2.75 h measurement.**

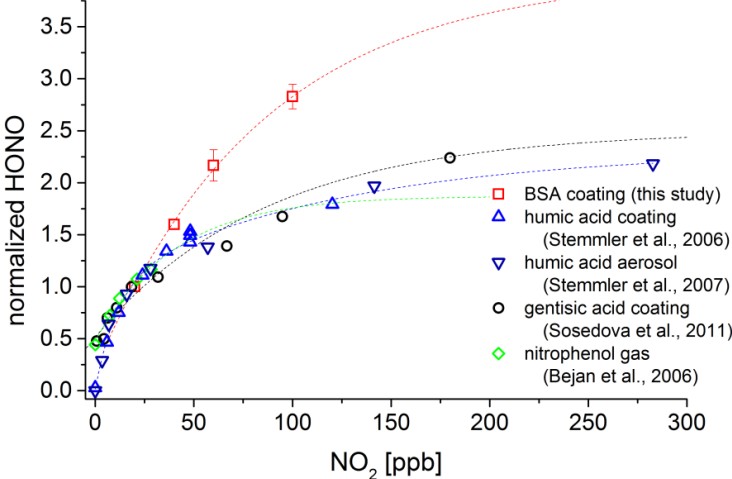

**Fig. 5: Comparison of HONO formation dependency on NO$_2$ at different organic surfaces. HONO concentrations are**
**normalized to the HONO concentration at 20 ppb NO$_2$ ([HONO]$_{NO2}$/[HONO]$_{NO2=20ppb}$). Red square = BSA coating (16 µg**
**cm$^{-2}$) at 161 W m$^{-2}$ and 50% RH (this study), blue triangles pointing up = humic acid coating (8 µg cm$^{-2}$) at 162 W m$^{-2}$ and**
**20% RH (Stemmler et al., 2006), dark blue triangles pointing down = humic acid aerosol with 100 nm diameter and a**
**surface of 0.151 m$^2$ m$^{-3}$ at 26% RH and 1x10$^{17}$ photons cm$^{-2}$ s$^{-1}$ (Stemmler et al., 2007), black circles = gentisic acid coating**
**(160-200 µg cm$^{-2}$) at 40-45% RH and light intensity similar as in the humic acid aerosol case (Sosedova et al., 2011), green**
**diamonds = ortho-nitrophenol in gas phase (ppm level) illuminated with UV/VIS light.**





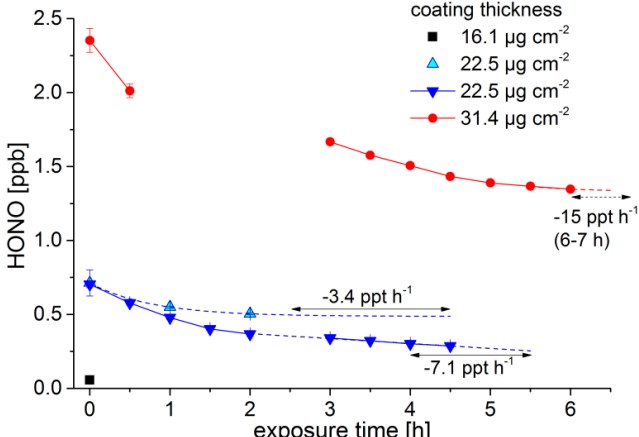

**Fig. 6: HONO formation on three different BSA coating thicknesses, exposed to 20 ppb of NO₂ under illuminated**
**conditions (7 VIS lamps). The HONO concentrations were normalized to reaction tube coverage (black: 100% of reaction**
**tube was covered with BSA, blueish: 70% of tube was covered and red: 50% of tube was covered with BSA). The middle**
**thick coating (22.46 μg cm⁻²) was replicated and studied with different reaction times (cyan and blue triangle). Solid lines**
**(with circles or triangles) present continuous measurements, when those are interrupted other conditions (e.g. light**
**intensity, NO₂ concentration) prevailed. Dotted lines show interpolations. Arrows indicate the intervals in which the**
**shown decay rates were determined. Error bars indicates standard deviations from 10-20 measuring points (5-10 min).**

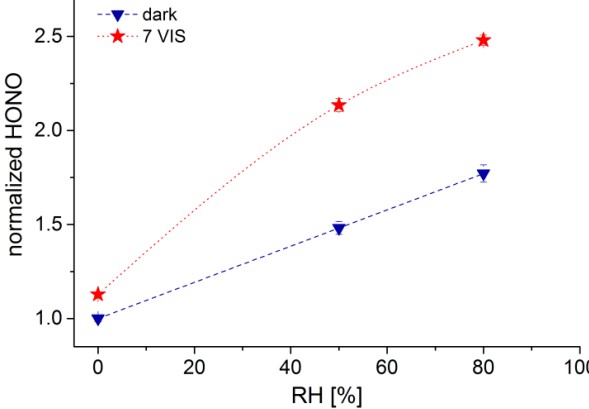

**Fig. 7: Dependency of humidity on the transformation of 25 ppb NO₂ in darkness (blue triangle) and with 7 VIS lights (red**
**star). HONO was normalized to HONO concentrations in darkness under dry conditions**
**([HONO]lights on-off; RH/[HONO]dark; RH=0).**



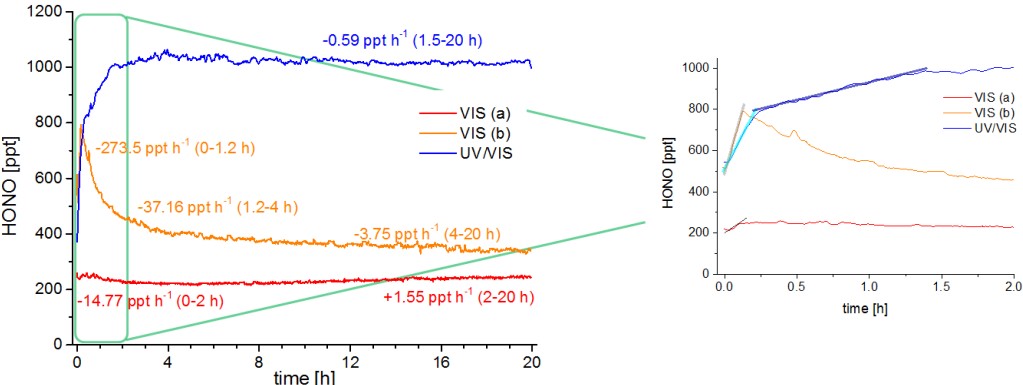

**Fig. 8: Extended (20 h) measurements of light-enhanced HONO formation on BSA (three coatings of 17.5 µg cm⁻²) at 80%**
**RH, 100 ppb NO₂. HONO decay rates [ppt h⁻¹] are shown with time periods (in brackets) in which they were calculated,**
**suggesting a stable HONO formation after 4 hours. Right: zoom in on the first 2 hours. Straight lines (black, grey, light**
**and dark blue) show the regressions of which d[HONO]/dt were used in the kinetic studies.**

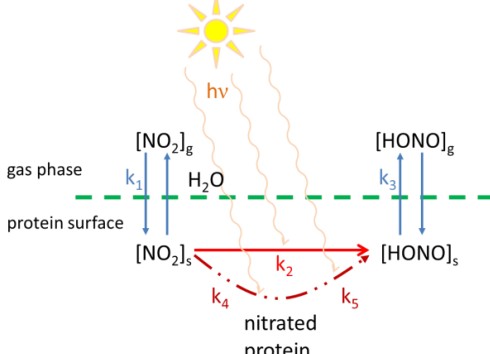

**Fig. 9: Schematic illustration of the underlying Langmuir-Hinshelwood-mechanism of light induced HONO formation on**
**protein surface.**