# Peer review of "Light-induced protein nitration and degradation with HONO 1"

_Atmospheric Chemistry and Physics, 2017_

## Referee Comment (RC1) · Anonymous Referee #1 · 30 Apr 2017

General Comments. This manuscript reports results of a study aimed at investigating photochemical formation of HONO from proteins exposed to NO2. The study employs coated wall flow tube techniques with LOPAP detection of HONO and chemiluminescence detection of NO2. The methods are appropriate for such a study and the results appear to meet the standards required by ACP. The topic is important as it addresses the byproducts associated with light-induced nitration of protein aerosols (e.g., pollen and other biological aerosols); it is novel in that it attempts to address the photochemical fate of the nitrated proteins. The relevance of protein nitration to the potency of allergens has been discussed in several publications, so that is clear. However, it is not so clear that nitrated proteins will be an important component of the daytime HONO budget since proteinaceous aerosols would constitute only a minor fraction of the total aerosol surface area in the atmosphere. Furthermore, strong evidence has

recently surfaced showing that the daytime HONO source is not linked to NO2 (see Pusede et al. Environ. Sci. Technol. 2016). In addition, there are limited situations where the aerosol phase has proved to have an impact on atmospheric HONO concentrations. Perhaps the authors could add a more extensive discussion of settings where they predict this chemistry to be important? Regardless, it is my opinion that the chemistry presented is interesting enough to warrant publication after these issues are addressed.

Specific Comments.

Page 1, line 20: The authors write that "nitration degrees of about 1% were derived applying NO2 concentrations..." How was the nitration degree determined?

1, 21: The term "Gas exchange measurements of TNM-nitrated proteins" is ambiguous.

1, 23: The term "fumigation" is not appropriate here. Please replace.

3, 22-24: I note that nitrated ovalbumin (OVA) was used in only one experiment in this study (section 3.2.1) while bovine serum albumin (BSA) was used for everything else. Ideally, one would use one protein for all the studies to facilitate comparison of results. Please explain why one protein was not used for everything.

3, 32: The methods section indicates that tetranitromethane is used to nitrate the OVA samples. This is a highly toxic and explosive reagent. Appropriate warnings should be included in this section to bring awareness of the dangers of using this reagent to anyone wishing to repeat these experiments.

9, 33 (and other places in the text, e.g. 10, 4): The term "catalytic converter" is an engineering term and is not appropriate in this context. I would replace with "catalytic surface".

10, 6: It is not clear what ND refers to in this line. Please clarify.

10, 27: It seems to me the term [HONO]1 + [HONO]2 is incorrect. Instead of indicating

concentrations, should one not be using rates (i.e., d[HONO]1/dt + d[HONO]2/dt)?

10, Kinetics studies section: The derivation of some of the indicated terms is not so clear. I question the need to go into the level of detail displayed in eq. 1-5. Please check over the derivation of keff. Also, perhaps I missed this explanation, but why are the reversible reactions in Figure 9 not included?

Figure 1: Ozone is included above the arrow in the first step. However, there is no indication that ozone was used in this study. Please clarify or correct.

---

## Referee Comment (RC2) · Anonymous Referee #2 · 1 May 2017

Overview

In this paper, titled "Light-induced protein nitration and degradation with HONO emission" by Meusel et al., the authors present an interesting dataset focused on the uptake of NO2 and subsequent emission of HONO by protein surfaces. HONO is an important reservoir for OH radicals and NOx, but very little is known about its formation and subsequent photochemistry on the surface of aerosol particles, which represent a significant amount of reactive surface area in the atmosphere. Therefore, the topic is very much atmospherically relevant. Based on a series of flow tube experiments, the authors find a dependence of NO2 uptake and subsequent emission of HONO on light intensity, relative humidity, NO2 concentration, and flow tube coating thickness. The authors argue that surface-enhanced NO2 conversion to HONO follows a Langmuir-Hinshelwood reaction mechanism. While I find the topic to be of general interest to

the community, I have several concerns regarding the experimental approach and interpretation, and therefore request that the authors make significant revisions to their manuscript before publication in ACP after considering my comments listed below.

General Comments

1. Section 3.1 (lines 22-23): The authors indicate that additional continuous exposure of the protein surface by light fully decomposed the protein so that no intact protein could be detected. However, the authors should clarify if only the nitrated protein residues decompose or all (nitrated and non-nitrated), and how that might affect ND.

2. Could the authors discuss the atmospheric implications of the irradiance intensity applied in this study compared to the solar irradiance intensity? They mention that their irradiance was 40% of clear sky conditions, similar to cloudy days, so does that imply that this chemistry could be more relevant in the atmosphere than the results suggest? Please elaborate.

3. In the VIS light wavelength range of the lamps used in this study (between 400 nm and 700 nm), NO2 photolysis could be significant and play an important role in the degree of protein nitration and HONO production. Was NO2 photolysis a concern and how might it affect the results?

4. In the last paragraph of the results section 3.1, the authors compare their results, which were conducted in the presence of NO2, with other nitration studies conducted in the presence of both NO2 and O3. How are these comparable, since NO2 and O3 combine to make N2O5 and NO3, which is a much more effective nitrating agent? The authors argue that their low ND may be due to light exposure, whereas the studies with larger ND that they compare to were conducted in the dark in the presence of NO3, so wouldn't the authors expect more ND in the other studies anyway because of the higher reactivity of NO3?

5. Section 3.2.4: The authors conclude that HONO production is greater for larger protein coating thicknesses. However, the coatings also covered different surface area of the flow tube. Do you expect surface area to be important in the context of this study? My concern is that by shortening the coated length of the flow tube for the thicker coating experiments, the authors potentially introduce bias in their measurement since both NO2 and HONO are exposed to different coated surface areas of the flow tube. Following NO2 uptake by the shorter coated length flow tube, the HONO that is emitted is subsequently exposed to less coated surface area for the remaining length of the flow tube. If a fraction of the HONO is taken up by the protein surface, less protein surface area implies more of the HONO is present in the gas phase. A better approach would have been to either maintain the same length of coated flow tube between experiments or to maintain the same surface concentration of protein between experiments for different coated lengths. The authors should at least discuss potential caveats for changing the coated surface area of the flow tube between experiments.

6. The rate of HONO emission decay as a function of exposure time as presented in Fig. 6 is also a bit confusing; the authors report emission decay rates in the range of 10-20 ppt hr-1, but it is difficult to tell from the y-axis since [HONO] is reported in ppb. It would help if the y-axis and reported rates had the same concentration units. The authors might also consider changing their y-axis to a log scale or plotting the red data points on a separate y axis, so the reader can better observe the decay for different time periods. However, it appears the rate is more on the order of 160 ppt hr-1 (linearly interpolated between 0 and 3 hrs). Why were the HONO emission decay rates only reported near the end of the exposure period (assuming the reported rates cover the exposure periods indicated by the arrows in Fig. 6)?

7. Given the apparent strong dependence on coating thickness, how relevant are the thicknesses of the coatings applied to the flow tube (>200 nm) compared to typical atmospheric aerosol? The authors should at least discuss the implications of coating thickness and HONO formation in the context of atmospheric aerosol particles.

8. Section 3.2.6: Have the authors considered to what extent photolysis of HONO

(in the case of the UV/VIS experiment) plays in the temporal evolution of the HONO concentration? The authors argue that the plateau in the HONO concentration in Fig. 8, followed by continuous and relatively stable emission of HONO from the protein surface is consistent with a Langmuir-Hinshelwood reaction mechanism. However, HONO photolyzes under UV conditions (300 nm < $\lambda$ < 400 nm), so might there be a point when the temporal HONO emission profile becomes limited by photolysis? The authors might consider including a photolysis term in their kinetics calculation (for both NO2 and HONO), e.g. d[NO2]s/dt = k1×[NO2]g – j(NO2)×[NO2]g and d[HONO]g/dt = k3×[HONO]s – j(HONO)×[HONO]g.

9. Section 3.3 and Fig. 8: Here, it appears the authors apply a series of kinetic equations to describe the temporal HONO emission profile shown in Fig. 8 based on Langmuir-Hinshelwood reaction kinetics. First, it is unclear if the lines plotted on top of the "UV/VIS" blue line in Fig. 8 are actually based on the kinetic equations described in section 3.3 or if they are simply linear fits with no theoretical basis, because in the figure description it states, "Straight lines. . .show the regressions. . ." If they are simply linear fits and then the kinetic terms were derived from the linear regression, my concern is this introduces significant ambiguity in the derived kinetics terms, because then the choice for each modeled section is entirely dependent on the user and not based on a sound theoretical description. Please clarify in both the Fig. 8 description and in sec. 3.3 whether these are simply linear fits or modeled based on the kinetic equations described in sec. 3.3. Furthermore, the authors must clarify what values were used (or derived from the linear fit) for k1, k2, k3, k4, k5, and k'. As a sensitivity test and validation of their model, I encourage the authors to apply their derived kinetic terms to model [HONO] as a function of [NO2], as shown in Fig. 5. Can [HONO] as a function of [NO2] be reproduced from the Langmuir–Hinshelwood terms described in sec. 3.3? Regarding Fig. 5, what do the dashed lines represent, are they fits to the data or just there to guide the eye? Please clarify in the figure description. Alternatively, the authors could plot their derived uptake coefficients (instead of [HONO]) as a function of time, and apply the Langmuir–Hinshelwood framework, e.g., as described in Ammann et al.

[Figure]

[2003]. This would also enable derivation of key kinetic terms describing NO2 uptake by proteinaceous aerosol surfaces, including the Langmuir equilibrium constant, surface accommodation coefficient and second-order surface reaction rate constant, which the community might find useful.

10. Have the authors considered the impact of photolysis of adsorbed HNO3 on the production of HONO in this study?

HNO3(ads) + h$\nu$ –> HONO + O

Given the high relative humidity and [NO2], HNO3 adsorption or formation on the surface of the flow tube could be substantial. While there was some mention in the introduction that HONO production from the photolysis of HNO3 may be important on organic substrates and soot, it was not discussed in the context of this study. The authors might consider estimating the contribution of adsorbed HNO3 photolysis to HONO produced in their flow tube experiments. Adsorbed HNO3 could be estimated based on the applied relative humidity and [NO2] (and assuming some reasonable surface coverage of HNO3), and the photolysis rate of HNO3, e.g., as determined in a very recent study by Laufs and Kleffmann [2016].

Minor Comments

1. Page 6, lines 8-9: It's not clear what the authors mean by "condensing condition" at a relative humidity (RH) of 98%, but not at 92%? Does this mean that the protein undergoes deliquescence at RH=98% and not 92%?

2. Figure 6: Along with the surface concentration of the coating (in units of $\mu$g cm-2), please include the calculated thickness of the coating in units of nm.

3. Summary and conclusions section, page 11 line 34: What is the significance of 1m2 of BSA surface or how was that surface area chosen?

References

Ammann, M., U. Poschl, and Y. Rudich (2003), Effects of reversible adsorption and Langmuir-Hinshelwood surface reactions on gas uptake by atmospheric particles, Phys Chem Chem Phys, 5(2), 351-356.

Laufs, S., and J. Kleffmann (2016), Investigations on HONO formation from photolysis of adsorbed HNO3 on quartz glass surfaces, Phys. Chem. Chem. Phys., 18, 9616-9625.
* * *

---

## Referee Comment (RC3) · Anonymous Referee #3 · 8 May 2017

This MS reports on HONO formation resulting (mostly) from the interaction of NO2 with a particular protein under visible illumination in a flow tube reactor. The HONO released to the gas phase is formed both by photolysis of nitrated tyrosine and a Langmuir-Hinshelwood surface reaction involving NO2 uptake; this latter process forms HONO even in the dark. For bot dark and illuminated channels, there is a positive dependence on RH which suggests that water is involved somehow, although this may be by changing the protein surface morphology rather than as a chemical promoter.

The experiments are well constructed and the results are of some interest. I do have a few comments for the authors' consideration, however:

1. page 5, lines 28-29: I am not convinced that you have demonstrated nitration with the very small signal reported.

2. page 6, lines 3-5: Again, this is one possible inference, but is certainly not conclusively shown!

3. page 6, section 3.2.1: this experiment is very poorly described - please explain exactly what was done.

4. Page 6, line 32-33: Could this be related to the photodecomposition of the protein, reposted above?

5. Sections 3.2.2 and 3.2.3: Brigante et al (J. Phys. Chem. A 2008, 112, 9503–9508) made these same observations.

6. Page 8, line 19-20: On what basis do you claim that nitration / reaction takes place below the surface layer?

7. page 8, line 28, ff: Brigante et al (2008) also saw no RH dependence for NO2/HONO on solid pyrene.

8. page 10, Eq. 1 and kinetic arguments: Why is the desorption reaction not included here? The implication of the L-H mechanism, suggested in Fig 5, is that this should be important. The kinetic scheme should reflect this, I think.

9. page 11, lines 17-23: This paragraph seems out of place here; perhaps in the Conclusions? In its place - can the authors in any way (semi)quantify their suggestion that HONO production via NO2/protein interactions could be atmospherically important?

10. The figure captions are not very descriptive. They should be rewritten, to explain what is displayed in the figures.
* * *

---

## Author Comment (AC1) · 9 Aug 2017

**General Comments**.
*This manuscript reports results of a study aimed at investigating photochemical formation of HONO from proteins exposed to NO2. The study employs coated wall flow tube techniques with LOPAP detection of HONO and chemiluminescence detection of NO2. The methods are appropriate for such a study and the results appear to meet the standards required by ACP. The topic is important as it addresses the byproducts associated with light-induced nitration of protein aerosols (e.g., pollen and other biological aerosols); it is novel in that it attempts to address the photochemical fate of the nitrated proteins.*

*The relevance of protein nitration to the potency of allergens has been discussed in several publications, so that is clear. However, it is not so clear that nitrated proteins will be an important component of the daytime HONO budget since proteinaceous aerosols would constitute only a minor fraction of the total aerosol surface area in the atmosphere. Furthermore, strong evidence has recently surfaced showing that the daytime HONO source is not linked to NO2 (see Pusede et al. Environ. Sci. Technol. 2016). In addition, there are limited situations where the aerosol phase has proved to have an impact on atmospheric HONO concentrations. Perhaps the authors could add a more extensive discussion of settings where they predict this chemistry to be important? Regardless, it is my opinion that the chemistry presented is interesting enough to warrant publication after these issues are addressed.*

**Response:**
Pusede et al., 2015 (Environ. Sci. Technol., 49, 12774-12781, 2015) observed no significant weekday-weekend difference in HONO levels during daytime, while $NO_2$ levels changed significantly during weekday-weekend and concluded that HONO didn´t derive from $NO_2$. Several studies didn´t find correlations with $NO_2$, but much more publications see a correlation with $NO_2$ and an enhanced correlation with $NO_2*J$ (e.g. Costabile et al., 2010; Spataro et al., 2013; Sörgel et al., 2011 + 2015; Su et al., 2008; Lee et al., 2016), both being in line with our observations presented in here. The absence of $NO_2$-HONO correlation does not exclude the involvement of $NO_2$ conversion in HONO. Whatever the detailed mechanism is, there are many complex processes involved in aerosol particles and on the ground surface that could lead to a highly non-linear dependence on $NO_2$ in both concentration and time domains (HONO precursors may be stored in reservoirs, both in the physical and chemical senses). Besides heterogeneous photochemistry on aerosols also heterogeneous photoenhanced $NO_2$ conversion on ground surfaces has been proposed (Ren et al., 2011; Laufs et al., 2017). As proteins are found in both aerosol particles (coarse and fine mode) as well as on most ground surfaces (soil, leaf etc.), we think that their widespread occurrence provides reasonable justification to have a closer look into the characteristics of their HONO emissions. Indeed, we agree with the referee that pinning down their impact in individual settings is crucial, but for the time being, too uncertain to make a strong statement in here.

**Comment:**

*Page 1, line 20: The authors write that "nitration degrees of about 1% were derived applying NO2 concentrations…" How was the nitration degree determined?*

**Response:**

The nitration degree is defined as the concentration of nitrated tyrosine divided by the concentration of all tyrosine residues. As it is written in the method part (2.1.), the nitration degrees were determined by HPLC-DAD analysis. Nitrated tyrosine residues were detected at 357 nm (and 280 nm) while tyrosine residues were detected at 280 nm only.

The respective section in the method part of the manuscript (page 4 lines 13-15) was slightly modified:

"Absorbance was monitored at wavelengths of 280 (tyrosine) and 357 nm (nitrotyrosine). The sample injection volume was 10-30 μL. Each chromatographic run was repeated three times. The protein nitration degree, which is defined as the ratio of nitrated tyrosine to all tyrosine residues, was determined by the method of Selzle et al. (2013). Native and un-treated BSA did not show any degree of nitration. "

**Comment:**

*1, 21: The term "Gas exchange measurements of TNM-nitrated proteins" is ambiguous.*

**Response:**

Now corrected in the manuscript (page 1, lines 21-22) to: "Measurements of gas exchange on TNM-nitrated proteins…"

**Comment:**

*1, 23: The term "fumigation" is not appropriate here. Please replace.*

**Response:**

Now corrected to "$NO_2$ exposure…"

**Comment:**

*3, 22-24: I note that nitrated ovalbumin (OVA) was used in only one experiment in this study (section 3.2.1) while bovine serum albumin (BSA) was used for everything else. Ideally, one would use one protein for all the studies to facilitate comparison of results. Please explain why one protein was not used for everything.*

**Response:**

Unfortunately, only nitrated OVA but no nitrated BSA was available from our partner.

**Comment:**

*3, 32: The methods section indicates that tetranitromethane is used to nitrate the OVA samples. This is a highly toxic and explosive reagent. Appropriate warnings should be included in this section to bring awareness of the dangers of using this reagent to anyone wishing to repeat these experiments.*

**Response:**

Although in most other publications safety notes/warnings of toxic chemicals are not mentioned, we acknowledge the advice and now added a respective note in the manuscript (page 3, line 34-35):

 "Please note that TNM is toxic if swallowed, can cause skin, eye and respiration irritation, is suspected to cause cancer and causes fire or explosion."

**Comment:**

*9, 33 (and other places in the text, e.g. 10, 14): The term "catalytic converter" is an engineering term and is not appropriate in this context. I would replace with "catalytic surface".*

**Response:**

Now corrected in the text according to the referee's suggestion.

**Comment:**

*10, 6: It is not clear what ND refers to in this line. Please clarify.*

**Response:**

ND refers for nitration degree as in the whole manuscript. This abbreviation was introduced when first mentioned in the manuscript on page 5 lines 19/20 (original manuscript). The nitration degree is defined as the concentration of nitrated tyrosine divided by the concentration of all tyrosine residues (see also comment above).

**Comment:**

*10, 27: It seems to me the term [HONO]1 + [HONO]2 is incorrect. Instead of indicating concentrations, should one not be using rates (i.e., d[HONO]1/dt + d[HONO]2/dt)?*

**Response:**

Thank you very much for noting, indeed that's the case. Now corrected in the text as suggested by the referee and moved to a new supplement (see comment below).

**Comment:**

*Kinetics studies section: The derivation of some of the indicated terms is not so clear. I question the need to go into the level of detail displayed in eq. 1-5. Please check over the derivation of keff. Also, perhaps I missed this explanation, but why are the reversible reactions in Figure 9 not included?*

**Response:**

To simplify the calculations, the reversible processes of $NO_2$ were neglected ($k_1$ would be the effective rate constant for the adsorption; including adsorption and desorption). In addition, the adsorption of HONO to the protein surface is supposed to be very small in relation to the desorption, as proteins are slightly acidic (similar to $k_1$ $k_3$ would be the effective rate constant including desorption and adsorption). Details are now displayed and discussed in a new supplement.

**Comment:**

*Figure 1: Ozone is included above the arrow in the first step. However, there is no indication that ozone was used in this study. Please clarify or correct.*

**Response:**

True, in our study only $N_2$ was applied as a carrier gas (no $O_3$). Fig. 1 is meant to give a complete overview of possible nitration mechanisms and refers to another study on nitration of proteins with $O_3$ and $NO_2$ (Shiraiwa et al., 2012 as indicated in the caption).

Figure 1 caption was modified: "Overview on possible reaction mechanism..."

**Reference:**

Costabile, F., Amoroso, A., and Wang, F.: Sub-mu m particle size distributions in a suburban Mediterranean area. Aerosol populations and their possible relationship with HONO mixing ratios, Atmospheric Environment, 44, 5258-5268, 10.1016/j.atmosenv.2010.08.018, 2010.

Laufs, S., Cazaunau, M., Stella, P., Kurtenbach, R., Cellier, P., Mellouki, A., Loubet, B., and Kleffmann, J.: Diurnal fluxes of HONO above a crop rotation, Atmos. Chem. Phys., 17, 6907-6923, 10.5194/acp-17-6907-2017, 2017.

Lee, J. D., Whalley, L. K., Heard, D. E., Stone, D., Dunmore, R. E., Hamilton, J. F., Young, D. E., Allan, J. D., Laufs, S., and Kleffmann, J.: Detailed budget analysis of HONO in central London reveals a missing daytime source, Atmos. Chem. Phys., 16, 2747-2764, 10.5194/acp-16-2747-2016, 2016.

Pusede, S. E., VandenBoer, T. C., Murphy, J. G., Markovic, M. Z., Young, C. J., Veres, P. R., Roberts, J. M., Washenfelder, R. A., Brown, S. S., Ren, X., Tsai, C., Stutz, J., Brune, W. H., Browne, E. C., Wooldridge, P. J., Graham, A. R., Weber, R., Goldstein, A. H., Dusanter, S., Griffith, S. M., Stevens, P. S., Lefer, B. L., and Cohen, R. C.: An Atmospheric Constraint on the NO2 Dependence of Daytime Near-Surface Nitrous Acid (HONO), Environmental Science & Technology, 49, 12774-12781, 10.1021/acs.est.5b02511, 2015.

Ren, X., Sanders, J. E., Rajendran, A., Weber, R. J., Goldstein, A. H., Pusede, S. E., Browne, E. C., Min, K. E., and Cohen, R. C.: A relaxed eddy accumulation system for measuring vertical fluxes of nitrous acid, Atmospheric Measurement Techniques, 4, 2093-2103, 10.5194/amt-4-2093-2011, 2011.

Selzle, K., Ackaert, C., Kampf, C. J., Kunert, A. T., Duschl, A., Oostingh, G. J., and Poschl, U.: Determination of nitration degrees for the birch pollen allergen Bet v 1, Analytical and Bioanalytical Chemistry, 405, 8945-8949, 10.1007/s00216-013-7324-0, 2013.

Shiraiwa, M., Selzle, K., Yang, H., Sosedova, Y., Ammann, M., and Poeschl, U.: Multiphase Chemical Kinetics of the Nitration of Aerosolized Protein by Ozone and Nitrogen Dioxide, Environmental Science & Technology, 46, 6672-6680, 10.1021/es300871b, 2012.

Sörgel, M., Regelin, E., Bozem, H., Diesch, J. M., Drewnick, F., Fischer, H., Harder, H., Held, A., Hosaynali-Beygi, Z., Martinez, M., and Zetzsch, C.: Quantification of the unknown HONO daytime source and its relation to NO2, Atmospheric Chemistry and Physics, 11, 10433-10447, 10.5194/acp-11-10433-2011, 2011.

Sörgel, M., Trebs, I., Wu, D., and Held, A.: A comparison of measured HONO uptake and release with calculated source strengths in a heterogeneous forest environment, Atmos. Chem. Phys., 15, 9237-9251, 10.5194/acp-15-9237-2015, 2015.

Spataro, F., Ianniello, A., Esposito, G., Allegrini, I., Zhu, T., and Hu, M.: Occurrence of atmospheric nitrous acid in the urban area of Beijing (China), The Science of the total environment, 447, 210-224, 10.1016/j.scitotenv.2012.12.065, 2013.

Su, H., Cheng, Y. F., Shao, M., Gao, D. F., Yu, Z. Y., Zeng, L. M., Slanina, J., Zhang, Y. H., and Wiedensohler, A.: Nitrous acid (HONO) and its daytime sources at a rural site during the 2004 PRIDE-PRD experiment in China, Journal of Geophysical Research-Atmospheres, 113, 10.1029/2007jd009060, 2008.

---

## Author Comment (AC2) · 9 Aug 2017

*This MS reports on HONO formation resulting (mostly) from the interaction of NO2 with a particular protein under visible illumination in a flow tube reactor. The HONO released to the gas phase is formed both by photolysis of nitrated tyrosine and a Langmuir-Hinshelwood surface reaction involving NO2 uptake; this latter process forms HONO even in the dark. For bot dark and illuminated channels, there is a positive dependence on RH which suggests that water is involved somehow, although this may be by changing the protein surface morphology rather than as a chemical promoter. The experiments are well constructed and the results are of some interest. I do have a few comments for the authors' consideration, however:*

**Comment:**
*page 5, lines 28-29: I am not convinced that you have demonstrated nitration with the very small signal reported.*

**Response:**
The nitration degree was determined by HPLC-DAD as described elsewhere (Selzle et al., 2013). This technique is sensitive and well established (detection limit < 1%). The difference of the nitration degree of native BSA (ND = 0%) and BSA treated with $NO_2$/light (ND = 1%) is significant. Yes, it is a small nitration degree, but still nitration was detected!
Now modified in the main text (page 5 lines 25-26): "…nitration degree…by means of HPLC-DAD was (1.0±0.1)%., significantly higher than the ND of untreated BSA (0%)"

**Comment:**
*page 6, lines 3-5: Again, this is one possible inference, but is certainly not conclusively shown!*

**Response:**
We tune down the tone and it now reads, "…possibly *suggesting the deficiency…*"

**Comment:**
*page 6, section 3.2.1: this experiment is very poorly described - please explain exactly what was done.*

**Response:**
The method part (2.1 and 2.2) describes the procedure of the experiments and gives an overview on conditions… (table 1). In this experiment previously nitrated OVA (method part) was coated on a tube and irradiated with light (0,1,3,7 lights) while flushing with either zero air or 20 ppb $NO_2$. HONO emissions

were detected at the outlet. After the trace gas exchange measurements the protein was extracted by pure H2O and nitration degree was determined via HPLC-DAD.

Now modified in the main text: "To study HONO emission from nitrated proteins, OVA was nitrated with TNM (see section 2.1) in liquid phase. The nitrated OVA (2 mg; ND = 12.5%) was coated onto the reaction tube and exposed to VIS lights under either pure nitrogen flow or 20 ppb $NO_2$ gas. Strong HONO emissions were found…"

**Comment:**

*Page 6, line 32-33: Could this be related to the photodecomposition of the protein, reposted above?*

**Response:**

Yes indeed, as it was already stated in the main text: "…which is in line with the observed decomposition of the native protein presented above."

**Comment:**

*Sections 3.2.2 and 3.2.3: Brigante et al (J. Phys. Chem. A 2008, 112, 9503–9508) made these same observations.*

**Response:**

Brigante et al., (2008) observed a linear dependency of $NO_2$ loss (ln c0/c) to light intensity (number of photons) for the $NO_2$ uptake on pyrene. Furthermore, they plotted $NO_2$ uptake coefficient as function of $NO_2$ concentration and shows an exponential (decay) dependence. They found that roughly 15% of the $NO_2$ loss on pyrene accounts for HONO production. Both cannot be directly compared to our results ("saturation" of HONO formation at high light intensities and very high $NO_2$ concentration).

However, Brigante is now additionally cited when discussing similarities to other studies.

**Comment:**

*Page 8, line 19-20: On what basis do you claim that nitration / reaction takes place below the surface layer?*

**Response:**

Indeed, the dependency of layer thickness on the HONO formation is a complex matter. Light penetrates into the bulk (according to the set-up illumination is from outside - light will first pass the protein layer at the inner glass surface and then the layer in contact with the carrier gas) and hence activation of the aromatic residues of the protein and photolysis of nitrated proteins can occur in the bulk. Also $NO_2$ might diffuse into the bulk (depending on humidity and therefore viscosity/solid or semi-solid state), and the formed HONO would also be able to diffuse out of the bulk. But we didn't mean to say that the reaction takes place only below the surface. Our point is that the observed dependence on the coating thickness suggests the Indeed, the dependency of layer thickness on the HONO formation is a complex matter. Light penetrates into the bulk (according to the set-up illumination is from outside - light will first pass the protein layer at the inner glass surface and then the layer in contact with the carrier gas) and hence activation of the aromatic residues of the protein involvement of bulk reactions and the reactions can happen in both, surface and bulk phase.

We added one more conclusively sentence to the manuscript: "The observed dependence on the coating thickness suggests the involvement of the bulk reactions, but the reactions can happen in both, surface and bulk phase."

**Comment:**

*page 8, line 28, ff: Brigante et al (2008) also saw no RH dependence for NO2/HONO on solid pyrene.*

**Response:**

Now additionally cited in the manuscript: "No impact of humidity on $NO_2$ uptake coefficients on pyrene was detected (Brigante et al., 2008)"

**Comment:**

*page 10, Eq. 1 and kinetic arguments: Why is the desorption reaction not included here? The implication of the L-H mechanism, suggested in Fig 5, is that this should be important. The kinetic scheme should reflect this, I think.*

**Response:**

To simplify the calculations, the reversible processes were neglected. In addition, the adsorption of HONO to the protein surface is supposed to be very small in relation to the desorption as proteins are slightly acidic (please see respective comments/reply of referee #1)

Modifications in the manuscript accordingly to referee #1: the equations of the single processes (eq.1-5) were removed to a new supplement and only the final equation is shown.

**Comment:**

*page 11, lines 17-23: This paragraph seems out of place here; perhaps in the Conclusions? In its place - can the authors in any way (semi)quantify their suggestion that HONO production via NO2/protein interactions could be atmospherically important?*

**Response:**

Paragraph moved to section 3.2.1 (page 7, lines 3-9)
See also referee 2 conclusion section (page 13, lines 10-17)

**Comment:**

*The figure captions are not very descriptive. They should be rewritten, to explain what is displayed in the figures.*

**Response:**

The figure captions were modified:
The term "normalized HONO" (several y-axes) was changed to "scaled HONO".

[revised manuscript text omitted]

---

## Author Comment (AC3) · 9 Aug 2017

**Overview**

*In this paper, titled "Light-induced protein nitration and degradation with HONO emission" by Meusel et al., the authors present an interesting dataset focused on the uptake of NO2 and subsequent emission of HONO by protein surfaces. HONO is an important reservoir for OH radicals and NOx, but very little is known about its formation and subsequent photochemistry on the surface of aerosol particles, which represent a significant amount of reactive surface area in the atmosphere. Therefore, the topic is very much atmospherically relevant. Based on a series of flow tube experiments, the authors find a dependence of NO2 uptake and subsequent emission of HONO on light intensity, relative humidity, NO2 concentration, and flow tube coating thickness. The authors argue that surface-enhanced NO2 conversion to HONO follows a Langmuir-Hinshelwood reaction mechanism. While I find the topic to be of general interest to the community, I have several concerns regarding the experimental approach and interpretation, and therefore request that the authors make significant revisions to their manuscript before publication in ACP after considering my comments listed below.*

**Comment:**

*Section 3.1 (lines 22-23): The authors indicate that additional continuous exposure of the protein surface by light fully decomposed the protein so that no intact protein could be detected. However, the authors should clarify if only the nitrated protein residues decompose or all (nitrated and non-nitrated), and how that might affect ND.*

**Response:**

The nitration degree in this study was detected with HPLC-DAD which only can detect the intact protein and the nitrated protein, but no possible degradation products like peptides, single amino acids and their nitrated forms, as those compounds are "filtered out" by the chromatography. According to our HPLC-DAD results also the non-nitrated proteins decomposed, i.e., the peak was below detection limit. If only nitrated-proteins decompose, the results would indicate that all proteins were firstly nitrated prior of being decomposed. Amino acids and peptides might still be present, either nitrated or not.

Now better specified in the main text of the manuscript (page 5, line 27): "Note that no intact protein (nitrated and non-nitrated) could be detected by HPLC-DAD after another 20 hours of irradiation without $NO_2$, indicating light induced decomposition of proteins"

**Comment:**

*Could the authors discuss the atmospheric implications of the irradiance intensity applied in this study compared to the solar irradiance intensity? They mention that their irradiance was 40% of clear sky conditions, similar to cloudy days, so does that imply that this chemistry could be more relevant in the atmosphere than the results suggest? Please elaborate.*

**Response:**

There are two different processes to discuss: 1) the degradation of nitrated proteins with HONO formation and 2) the heterogeneous $NO_2$ conversion. As shown in Fig.3 the light dependency of HONO formation from previously nitrated proteins is almost linear in the range of applied light intensity. So here, yes during sunny days, when irradiance is higher than our applied light intensity, an even higher HONO formation can be anticipated, accompanied by a faster degradation of the nitrated proteins. However, as the dependency for higher light intensities was not investigated, we cannot make a firm statement here. The observed light dependency of the heterogeneous conversion of $NO_2$ on BSA was not linear as shown in Fig. 4b. Here our results rather indicate an upper limit for the HONO formation, as was reported similarly by Stemmler et al. (2006, 2007).

Now added in the conclusion (page 13, line 4-7): "While heterogeneous HONO formation of BSA exposed to $NO_2$ revealed light saturation at intensities higher than 161 W m$^{-2}$, the HONO formation from previously nitrated OVA was linearly increasing over the whole light intensity range investigated. The latter let assume even higher HONO formation under sunny (clear sky) ambient atmospheric conditions."

**Comment:**

*In the VIS light wavelength range of the lamps used in this study (between 400 nm and 700 nm), NO2 photolysis could be significant and play an important role in the degree of protein nitration and HONO production. Was NO2 photolysis a concern and how might it affect the results?*

**Response:**

Direct $NO_2$ photolysis (<400 nm) won´t occur under conditions applied in this study (Gardner et al., 1987; Roehl et al., 1994). There might be some electronic excitation of $NO_2$, which disproportionate to NO and $NO_3$. Stemmler et al., 2007, determined a photolysis frequency of $NO_2$ of up to 5 x 10$^{-4}$ s$^{-1}$ for very similar light conditions as we used, which is much lower than in the atmosphere (e.g. up to 1 x 10$^{-2}$ during the CYPHEX campaign 2014, Meusel et al., 2016). But $NO_3$ will probably directly deplete under this irradiation back to $NO_2$ or NO (Johnston et al., 1996).

$NO_2$ + hv $\rightarrow$ NO + O(3P) (< 400 nm)

$NO_2$ + hv $\rightarrow$ $NO_2$* (+$NO_2$) $\rightarrow$ $NO_3$ + NO (>410 nm)

$NO_3$ + hv $\rightarrow$ $NO_2$ + O(3P) (~580 nm)

$NO_3$ + hv $\rightarrow$ NO + $O_2$ (~ 600 nm)

Overall we assume the effect to be negligible. Furthermore, Shiraiwa et al. (2012) could exclude the importance of $NO_3$ (in their case formed by reaction of $NO_2$ + $O_3$) uptake on BSA.

Now added in section 3.1 (page 6, lines 16-23): "Shiraiwa et al. (2012) performed kinetic modelling and found that maximum 30% (conservative upper limit) of N-uptake on BSA could be explained by $NO_3$ or $N_2O_5$, which are generated by the reaction of $NO_2$ and $O_3$, while overall nitration was governed by an indirect mechanism in which a radical intermediate was formed by the reaction of BSA with ozone, which then reacted with $NO_2$. On NaCl surface N-uptake was dominated by $NO_3$ and $N_2O_5$. Furthermore, $NO_3$ radicals, which in this study could be formed by photolysis of $NO_2$ (>410 nm, disproportionation of excited $NO_2$), are not stable under the light condition applied (400-700 nm) (Johnston et al., 1996).

Therefore, in the present study reactions with $NO_3$ were neglected. Photolysis of $NO_2$ forming NO (< 400 nm) can also be neglected (Gardner et al., 1987; Roehl et al., 1994). A photolysis frequency for $NO_2$ of up to 5 x $10^{-4}$ $s^{-1}$ under similar experimental light conditions was determined by Stemmler et al., 2007."

**Comment:**

*In the last paragraph of the results section 3.1, the authors compare their results, which were conducted in the presence of NO2, with other nitration studies conducted in the presence of both NO2 and O3. How are these comparable, since NO2 and O3 combine to make N2O5 and NO3, which is a much more effective nitrating agent? The authors argue that their low ND may be due to light exposure, whereas the studies with larger ND that they compare to were conducted in the dark in the presence of NO3, so wouldn't the authors expect more ND in the other studies anyway because of the higher reactivity of NO3?*

**Response:**

Shiraiwa et al. (2012) estimated that maximum 30% of the N-uptake is due to $NO_3$ and $N_2O_5$ uptake on BSA, while overall nitration was governed by an indirect mechanism in which a radical intermediate was formed by the reaction of BSA with ozone, which then reacted with $NO_2$. On NaCl surfaces, on the other hand, $NO_3$ and $N_2O_5$ uptake dominate. Therefore, the higher ND of BSA exposed to $O_3$ and $NO_2$ is mainly due to higher activation by $O_3$ and due to BSA decomposition by light. Please also see the comment above.

**Comment:**

*Section 3.2.4: The authors conclude that HONO production is greater for larger protein coating thicknesses. However, the coatings also covered different surface area of the flow tube. Do you expect surface area to be important in the context of this study? My concern is that by shortening the coated length of the flow tube for the thicker coating experiments, the authors potentially introduce bias in their measurement since both NO2 and HONO are exposed to different coated surface areas of the flow tube. Following NO2 uptake by the shorter coated length flow tube, the HONO that is emitted is subsequently exposed to less coated surface area for the remaining length of the flow tube. If a fraction of the HONO is taken up by the protein surface, less protein surface area implies more of the HONO is present in the gas phase. A better approach would have been to either maintain the same length of coated flow tube between experiments or to maintain the same surface concentration of protein between experiments for different coated lengths. The authors should at least discuss potential caveats for changing the coated surface area of the flow tube between experiments.*

**Response:**

As we manually coated the reaction tube, it was difficult to obtain equally/consistent surfaces. Therefore, each coating was different and also the covered surface area could only be roughly estimated. So, we agree with the referee that the coating thickness/surface is the biggest uncertainty in the experiment. And yes, there might be a bias based on $NO_2$/HONO uptake/emission on/from different coating surface areas. But we expect that HONO uptake coefficients on both proteins (as slightly acidic) as well as on glass surfaces were small (Syomin and Finlayson-Pitts, 203), so that the difference of HONO uptake due to different surface areas and covered tube length is low. Also $NO_2$ uptake on glass is supposed to be significantly lower than on proteins. We don't expect a difference in tube coverage of 20% would increase HONO concentrations about three times.

According to the referee's suggestion, we now added in the main text (page 8, lines 19-24): "Exposing (20%) different coated surface areas in the flow tube, potentially introduced bias comparing different data sets. Emitted HONO might be re-adsorbed differently by proteins and glass surface. However, as the protein is slightly acidic, a low uptake efficiency of HONO by BSA can be anticipated, which should not differ too much from the un-covered glass tube surface (Syomin and Finlayson-Pitts, 2003). Accordingly, $NO_2$ uptake on glass is assumed to be significantly lower than on proteins."

**Comment:**

*The rate of HONO emission decay as a function of exposure time as presented in Fig. 6 is also a bit confusing; the authors report emission decay rates in the range of 10-20 ppt hr-1, but it is difficult to tell from the y-axis since [HONO] is reported in ppb. It would help if the y-axis and reported rates had the same concentration units. The authors might also consider changing their y-axis to a log scale or plotting the red data points on a separate y axis, so the reader can better observe the decay for different time periods. However, it appears the rate is more on the order of 160 ppt hr-1 (linearly interpolated between 0 and 3 hrs). Why were the HONO emission decay rates only reported near the end of the exposure period (assuming the reported rates cover the exposure periods indicated by the arrows in Fig. 6)?*

**Response:**

Fig. 6 does not show emission rates as a function of time, but normalized (to reaction tube coverage) HONO concentrations vs time! The numbers in the diagram represent the slope (decay rates) at the end (time period indicated by arrows), and indicate a stable HONO formation (as also seen in Fig.8). In figure 8 also several decay rates are shown for earlier exposure times, so that in the respective figure 6 the decay rates are only shown in the end when the concentrations are stable.

The unit of the y axis was changed from ppb to ppt.

**Comment:**

*Given the apparent strong dependence on coating thickness, how relevant are the thicknesses of the coatings applied to the flow tube (>200 nm) compared to typical atmospheric aerosol? The authors should at least discuss the implications of coating thickness and HONO formation in the context of atmospheric aerosol particles.*

**Response:**

Typical aerosol concentrations of bacteria, fungal spores and pollen are 0.1, 0.1-1 and 1 $\mu g\ m^{-3}$, respectively (Despres et al., 2012). Aerosol particles may contain up to 5% proteins. But it is not known how much proteins cover the aerosol surface nor how thick this coating would be. This was already mentioned in the conclusion part of the manuscript; and which is why it is hard to make a firm statement here. However, to address this important issue we now added in the text (page 13, lines10-17): "Typical aerosol surface concentrations in rural regions are about 100 $\mu m^2\ cm^{-3}$. Stemmler et al. (2007) estimated a HONO formation of 1.2 ppt $h^{-1}$ on pure humic acid aerosols in environmental conditions. As $NO_2$ uptake coefficients and HONO formation rates on proteins are similar to humic acid but only about 5% of the aerosol mass can be assumed to consist of proteins, it can be anticipated that HONO formation on aerosol is not a significant HONO source in ambient environmental settings. However, proteins on ground surfaces (soil, plants etc.) might play a more important role. Accordingly, Stemmler et al. (2006 and 2007) suggested that $NO_2$ conversion on soil covered with humic acid would be sufficient to explain missing HONO sources up to 700 ppt $h^{-1}$."

**Comment:**

*Section 3.2.6: Have the authors considered to what extent photolysis of HONO (in the case of the UV/VIS experiment) plays in the temporal evolution of the HONO concentration? The authors argue that the plateau in the HONO concentration in Fig. 8, followed by continuous and relatively stable emission of HONO from the protein surface is consistent with a Langmuir-Hinshelwood reaction mechanism. However, HONO photolyzes under UV conditions (300 nm < _ < 400 nm), so might there be a point when the temporal HONO emission profile becomes limited by photolysis? The authors might consider including a photolysis term in their kinetics calculation (for both NO2 and HONO), e.g. d[NO2]s/dt = k1_[NO2]g – j(NO2)_[NO2]g and d[HONO]g/dt =k3_[HONO]s – j(HONO)_[HONO]g.*

**Response:**

HONO photolysis was not considered. The overlap of UV light spectrum and HONO absorption/photolysis spectrum is quiet low about 340-400 nm. The quartz glass tube has a transmission of 90% at these wavelengths. The applied light intensity (with 7 lights on) is about 40% of a clear sky irradiance for a solar zenith of 48°. In clear sky HONO photolysis frequencies are in the range of 1.2-1.5 x $10^{-3}$ $s^{-1}$ (e.g., on Cyprus in summer 2014; Meusel et al. 2016). In the reaction tube the photolysis frequency would therefore decrease down to 0.4-0.5 x $10^{-3}$ $s^{-1}$. When only irradiated with VIS lights (exclusion of HONO photolysis, emission profile not limited) the pattern is the same as with UV (only a smaller absolute concentration) indicating a stable formation.

Now revised in the manuscript in the kinetic section (page 11, line 34 – page 12 line 3): "In this study, neither HONO nor $NO_2$ photolysis is considered, as the overlap of the applied UV/VIS or VIS range (340-700 nm or 400-700 nm) and the HONO and $NO_2$ photolysis spectrum (<400 nm) is low. Furthermore, the applied light intensity is lower compared to clear sky irradiance and the respective UV light is partly absorbed by the reaction tube although quartz glass was used (transmission ~ 90%) and the photolysis frequency would decrease down to $10^{-4}$ $s^{-1}$. Hence, the photolysis is assumed to be not significant."

**Comment:**

*Section 3.3 and Fig. 8: a) Here, it appears the authors apply a series of kinetic equations to describe the temporal HONO emission profile shown in Fig. 8 based on Langmuir-Hinshelwood reaction kinetics. First, it is unclear if the lines plotted on top of the "UV/VIS" blue line in Fig. 8 are actually based on the kinetic equations described in section 3.3 or if they are simply linear fits with no theoretical basis, because in the figure description it states, "Straight lines: : :show the regressions: : :" If they are simply linear fits and then the kinetic terms were derived from the linear regression, my concern is this introduces significant ambiguity in the derived kinetics terms, because then the choice for each modeled section is entirely dependent on the user and not based on a sound theoretical description. Please clarify in both the Fig. 8 description and in sec. 3.3 whether these are simply linear fits or modeled based on the kinetic equations described in sec. 3.3. Furthermore, the authors must clarify what values were used (or derived from the linear fit) for k1, k2, k3, k4, k5, and k'.*

*b) As a sensitivity test and validation of their model, I encourage the authors to apply their derived kinetic terms to model [HONO] as a function of [NO2], as shown in Fig. 5. Can [HONO] as a function of [NO2] be reproduced from the Langmuir–Hinshelwood terms described in sec. 3.3? Regarding Fig. 5, what do the dashed lines represent, are they fits to the data or just there to guide the eye? Please clarify in the figure description.*

*c) Alternatively, the authors could plot their derived uptake coefficients (instead of [HONO]) as a function of time, and apply the Langmuir–Hinshelwood framework, e.g., as described in Ammann et al. [2003].*

*This would also enable derivation of key kinetic terms describing NO2 uptake by proteinaceous aerosol surfaces, including the Langmuir equilibrium constant, surface accommodation coefficient and second-order surface reaction rate constant, which the community might find useful.*

**Response:**
a) The lines in fig 8 are linear fits (no theoretical basis). The slopes of those were taken to calculate $k_{eff}$. Other rate constants (k1, k2, k3, k4, k5, k') were not calculated. Single equations were moved to a new supplement.

b) In fig 5. the dotted lines are regressions of the measured data points (exponential fittigs, e.g., $y = y_0 + A * e^{-x/t}$) only to guide the eye (now better described in the figure captions).
In our kinetic study we calculated an effective rate constant for the $NO_2$ conversion on BSA. In a range of 0-100 ppb $NO_2$ the HONO formation is almost linear (fig 5), which would be also indicated by the Langmuir-Hinshelwood mechanism (here first rate: d[HONO]/dt = k*[NO$_2$]).

c) It is not possible to extract a full set of parameters for a LH model based on the present data. As pointed out in Bartels-Rausch et al. (2010) and to some degree also in the Stemmler et al. studies, the saturating behavior of photochemical HONO production may be due to either the adsorbed precursor on the surface or due to a photochemical competition process, which also leads to a Lindemann-Hinshelwood type kinetic expression (Minero, 1999). Therefore, mathematically, the rate expressions get a comparable NO2 pressure dependence. Therefore, measurements of the $NO_2$ dependence at different light intensities would be required to disentangle the two. The nearly single exponential (linear in the log-log plot) decay of gamma vs time in the figure below (fig. R1) indicates that the system is governed by degradation and not by reaction steady state, so that modelling the system explicitly in terms of all the kinetic parameters would be ambiguous.

[Figure]

Fig. R1: log-log plot of the uptake-coefficients vs time for the three different experiments, indicated with the different colors (from the long term and kinetic study)

Added to the manuscript (page 12, lines 24-28): "It was not possible to extract a set of parameters for a Langmuir Hinshelwood mechanism (like Langmuir equilibrium constant, surface accommodation coefficient or second order rate constant) from the presented data. The saturating behavior of photochemical HONO production may be due to either the adsorbed precursor on the surface or due to a photochemical competition process, which also leads to a Lindemann-Hinshelwood type kinetic expression (Minero, 1999)."

**Comment:**

*Have the authors considered the impact of photolysis of adsorbed HNO3 on the production of HONO in this study? HNO3(ads) + h_ –> HONO + O*

*Given the high relative humidity and [NO2], HNO3 adsorption or formation on the surface of the flow tube could be substantial. While there was some mention in the introduction that HONO production from the photolysis of HNO3 may be important on organic substrates and soot, it was not discussed in the context of this study. The authors might consider estimating the contribution of adsorbed HNO3 photolysis to HONO produced in their flow tube experiments. Adsorbed HNO3 could be estimated based on the applied relative humidity and [NO2] (and assuming some reasonable surface coverage of HNO3), and the photolysis rate of HNO3, e.g., as determined in a very recent study by Laufs and Kleffmann [2016].*

**Response:**

Gas phase reaction doesn't produce HNO3, because $N_2$, but not synthetic air was used as the carrier gas. Usually $HNO_3$ photolysis happens at < 350 nm. Photolysis of adsorbed $HNO_3$ might be shifted to slightly higher wavelength. In the publication of Laufs and Kleffmann (2016), J values ($HNO_3 \rightarrow HONO$) as low as $10^{-7}$ $s^{-1}$ were obtained. Our UV lamps had a spectral range of 340-400 nm. As a conclusion, $HNO_3$ photolysis was negligible in this study.

Added to the section 3.2.6(page 10, lines 18-23), as only here UV light was applied: "HONO formation by photolysis of (adsorbed) $HNO_3$ is assumed to be insignificant in this study. With $N_2$ as carrier gas, gas phase reactions of $NO_2$ do not produce $HNO_3$. Even when small amounts of $HNO_3$ would be formed by unknown heterogeneous reactions, photolysis of $HNO_3$ is only significant at wavelengths < 350 nm, which is close to the lowest limit of the UV wavelength applied in this study. Likewise, the respective photolysis frequency recently proposed by Laufs and Kleffmann (2016) of about 2.4 x $10^{-7}$ $s^{-1}$ is very low. "

**Minor Comment:**

*Page 6, lines 8-9: It's not clear what the authors mean by "condensing condition" at a relative humidity (RH) of 98%, but not at 92%? Does this mean that the protein undergoes deliquescence at RH=98% and not 92%?*

**Response:**

At 98% RH water vapor condensed (visible water layer), but not at lower RH of 92% (Reinmuth-Selzle et al., 2014). Deliquescence of BSA already occurs at 35% (see section 3.2.4).

**Minor Comment:**

*Figure 6: Along with the surface concentration of the coating (in units of _g cm-2), please include the calculated thickness of the coating in units of nm.*

**Response:**

According to the referee's advice, we now added the layer thickness in the plot:

[Figure]

Fig. R2 = new Fig. 6:  HONO concentration vs exposure time for different coating thicknesses.

**Minor Comment:**

Summary and conclusions section, page 11 line 34: What is the significance of 1m2 of BSA surface or how was that surface area chosen?

**Response:**

HONO formation per $m^2$ [ppt $h^{-1}$ $m^{-2}$]

Now the main text of the manuscript(page 13, lines 2-4) is rephrased to: "At 20 ppb $NO_2$ HONO formation of 19.8 ppb $h^{-1}$ $m^{-2}$ could be estimated"

**References**

Ammann, M., U. Poschl, and Y. Rudich: Effects of reversible adsorption and Langmuir-Hinshelwood surface reactions on gas uptake by atmospheric particles, Phys Chem Chem Phys, 5(2), 351-356, 2003.

Bartels-Rausch, T., Brigante, M., Elshorbany, Y. F., Ammann, M., D'Anna, B., George, C., Stemmler, K., Ndour, M., and Kleffmann, J.: Humic acid in ice: Photo-enhanced conversion of nitrogen dioxide into nitrous acid, Atmos. Environ., 44, 5443-5450, 2010.

Laufs, S., and J. Kleffmann: Investigations on HONO formation from photolysis of adsorbed HNO3 on quartz glass surfaces, Phys. Chem. Chem. Phys., 18, 9616-9625, 2016.

Després, V., Huffman, J. A., Burrows, S. M., Hoose, C., Safatov, A., Buryak, G., Fröhlich-Nowoisky, J., Elbert, W., Andreae, M., Pöschl, U., and Jaenicke, R.: Primary biological aerosol particles in the atmosphere: a review, Tellus B: Chemical and Physical Meteorology, 64, 15598, 10.3402/tellusb.v64i0.15598, 2012.

Gardner, E. P., Sperry, P. D., and Calvert, J. G.: Primary quantum yields of NO2 photodissociation, Journal of Geophysical Research: Atmospheres, 92, 6642-6652, 10.1029/JD092iD06p06642, 1987.

Johnston, H. S., Davis, H. F., and Lee, Y. T.: NO3 Photolysis Product Channels:  Quantum Yields from Observed Energy Thresholds, The Journal of Physical Chemistry, 100, 4713-4723, 10.1021/jp952692x, 1996.

Meusel, H., Kuhn, U., Reiffs, A., Mallik, C., Harder, H., Martinez, M., Schuladen, J., Bohn, B., Parchatka, U., Crowley, J. N., Fischer, H., Tomsche, L., Novelli, A., Hoffmann, T., Janssen, R. H. H., Hartogensis, O., Pikridas, M., Vrekoussis, M., Bourtsoukidis, E., Weber, B., Lelieveld, J., Williams, J., Pöschl, U., Cheng, Y., and Su, H.: Daytime formation of nitrous acid at a coastal remote site in Cyprus indicating a common ground source of atmospheric HONO and NO, Atmos. Chem. Phys., 16, 14475-14493, 10.5194/acp-16-14475-2016, 2016.

Minero, C.: Kinetic analysis of photoinduced reactions at the water semiconductor interface, Catal. Today, 54, 205-216, 1999.

Reinmuth-Selzle, K., Ackaert, C., Kampf, C. J., Samonig, M., Shiraiwa, M., Kofler, S., Yang, H., Gadermaier, G., Brandstetter, H., Huber, C. G., Duschl, A., Oostingh, G. J., and Pöschl, U.: Nitration of the Birch Pollen Allergen Bet v 1.0101: Efficiency and Site-Selectivity of Liquid and Gaseous Nitrating Agents, Journal of Proteome Research, 13, 1570-1577, 10.1021/pr401078h, 2014.

Roehl, C. M., Orlando, J. J., Tyndall, G. S., Shetter, R. E., Vazquez, G. J., Cantrell, C. A., and Calvert, J. G.: Temperature Dependence of the Quantum Yields for the Photolysis of NO2 Near the Dissociation Limit, The Journal of Physical Chemistry, 98, 7837-7843, 10.1021/j100083a015, 1994.

Shiraiwa, M., Selzle, K., Yang, H., Sosedova, Y., Ammann, M., and Poeschl, U.: Multiphase Chemical Kinetics of the Nitration of Aerosolized Protein by Ozone and Nitrogen Dioxide, Environmental Science & Technology, 46, 6672-6680, 10.1021/es300871b, 2012.

Syomin, D. A. and Finlayson-Pitts, B. J.: HONO decomposition on borosilicate glass surfaces: implications for environmental chamber studies and field experiments, Physical Chemistry Chemical Physics, 5, 5236-5242, 2003.

Stemmler, K., Ammann, M., Donders, C., Kleffmann, J., and George, C.: Photosensitized reduction of nitrogen dioxide on humic acid as a source of nitrous acid, Nature, 440, 195-198, 10.1038/nature04603, 2006.

Stemmler, K., Ndour, M., Elshorbany, Y., Kleffmann, J., D'Anna, B., George, C., Bohn, B., and Ammann, M.: Light induced conversion of nitrogen dioxide into nitrous acid on submicron humic acid aerosol, Atmospheric Chemistry and Physics, 7, 4237-4248, 2007.

---

## Referee Report (RR1)

**Review of the revised manuscript entitled "Light-induced protein nitration and degradation with HONO emission" by Hannah Meusel et al.**

I appreciate the author's careful consideration of my comments and the clarifications and changes to the manuscript. This has resulted in a much improved manuscript, in terms of content and readability. I recommend publication in ACP as is, but just note that the last comment of my original review actually concerned the following reaction, $NO_2 + H_2O + \text{surface} \rightarrow HONO + HNO_3$. The large mass accommodation coefficient and water solubility of $HNO_3$ imply it "sticks" to the flow tube walls and other surfaces, where the "adsorbed" $HNO_3$ could undergo photolysis and form HONO. Regardless, the wavelength range at which $HNO_3$ photolysis occurs is near the lower limit of the lamps used in this study and the photolysis rates of adsorbed $HNO_3$ reported in *Lauffs and Kauffmann* [2016]are relatively low.